# Ballistic Performance of Polyurea-Reinforced Ceramic/Metal Armor Subjected to Projectile Impact

**DOI:** 10.3390/ma15113918

**Published:** 2022-05-31

**Authors:** Peng Si, Yan Liu, Junbo Yan, Fan Bai, Fenglei Huang

**Affiliations:** 1State Key Laboratory of Explosion Science and Technology, Beijing Institute of Technology, Beijing 100081, China; bit_sipeng@163.com (P.S.); yanjunbo@bit.edu.cn (J.Y.); bitbaifan@bit.edu.cn (F.B.); huangfl@bit.edu.cn (F.H.); 2Beijing Institute of Technology Chongqing Innovation Center, Chongqing 401120, China

**Keywords:** ceramic armor, polyurea, ballistic performance, projectile impact, mass efficiency

## Abstract

Although polyurea has attracted extensive attention in impact mitigation due to its protective characteristics during intensive loading, the ballistic performance of polyurea-reinforced ceramic/metal armor remains unclear. In the present study, polyurea-reinforced ceramic/metal armor with different structures was designed, including three types of coating positions of the polyurea. The ballistic tests were conducted with a ballistic gun; the samples were subjected to a tungsten projectile formed into a cylinder 8 mm in diameter and 30 mm in length, and the deformation process of the tested targets was recorded with a high-speed camera. The ballistic performance of the polyurea-reinforced ceramic/metal armor was evaluated according to mass efficiency. The damaged targets were investigated in order to determine the failure patterns and the mechanisms of interaction between the projectile and the target. A scanning electron microscope (SEM) was used to observe the microstructure of polyurea and to understand its failure mechanisms. The results showed that the mass efficiency of the polyurea-coated armor was 89% higher than that of ceramic/metal armor, which implies that polyurea-coated ceramic armor achieved higher ballistic performance with lighter mass quality than that of ceramic/metal armor. The improvement of ballistic performance was due to the energy absorbed by polyurea during glass transition. These results are promising regarding further applications of polyurea-reinforced ceramic/metal armor.

## 1. Introduction

Ceramic armors have been extensively studied to develop new strategies to protect critical military equipment against the harmful effects of explosions and impacts. Thus, various ballistic protection systems have been developed. In the early stages, traditional armor systems on military vehicles had been made of monolithic metal materials, such as steel and aluminum. Recently, many new armor materials have been developed for use in military equipment, such as titanium alloys and metal–matrix composites [1]. Note that these materials provide adequate protection by increasing their thickness, but this strategy leads to poor fuel efficiency and loss of mobility. Therefore, several constraints must be considered when designing high-performance armor systems. For example, ballistic performance and being lightweight is crucial [2]. At a later stage, instead of monolithic metal armors, ceramic/metal armors were proposed to meet these requirements. These ceramic/metal armors are composed of ceramic on the strike face, with higher strength under compression than when under tension, and the metal is on the back, offsetting the drawbacks of ceramics [3]. The mechanism of ceramic/metal armors in protecting from projectile impact is that ceramics reduce the penetration ability of projectiles by deforming and eroding the projectile, then the backplate absorbs the residual kinetic energy [4,5,6].

Researchers have been exploring feasible approaches to improve the ballistic performance of ceramic armors. Previous studies investigated the influence of material properties on the ballistic performance of ceramic/metal armor. For example, the elastic modulus increases the interaction time between the projectile and the ceramic surface, thus allowing the ceramic to absorb more energy [7]. Additionally, there is a positive correlation between flexural strength/density ratio and ballistic resistance [8]. Nevertheless, the material properties analyzed in static conditions may not be directly translatable to the dynamic demands of ballistic performance [8,9,10]. In addition to their material properties, the ballistic performance of these materials can be significantly improved by establishing a reasonable structure [11,12,13,14,15]. For example, the ballistic limit can be increased by adjusting the thickness ratio and the restraint form of the ceramic block. Several studies have improved the ballistic performance by increasing the pre-stress of the ceramic [16,17,18,19]. Previous studies have also established theoretical models of ballistic limits for structure design to optimize ceramic/metal armor [20,21]. Meanwhile, the ballistic performance of composite armors containing, among other things, ceramic layers has been improved by introducing new materials, such as Kevlar and UHMWPE [22,23,24]. Such composite armors may achieve considerable ballistic resistance. However, the density of Kevlar fiber was higher than expected, and the melting point of UHMWPE was lower, which limited the applications of the two materials. Additionally, these materials may not fully meet the requirements of cost-effectiveness, low maintenance, and high performance needed for novel armor systems. Therefore, a composite armor containing novel materials is required.

A ceramic armor containing novel materials needs to be tested to offset the above deficiencies. Recently, polyurea has received extensive attention for its excellent performance in impact mitigation, due to its excellent mechanical properties [25,26,27,28,29]. The dynamic response of polyurea during impact exhibits strong rate-dependency [30,31,32]. With its glass transition during the impact, polyurea can reach a mass of energy dissipation that is correlated to the internal friction induced by deformation and the breakup of phase-segregated morphology [33]. It seems that the polyurea layer may achieve a compromise between ballistic-protection resistance and durability in ceramic composite armor by appropriately combining its viscoelastic and volumetric compressibility characteristics [34]. In addition, the coating position of polyurea is crucial, and its effect seems disparate among diverse substrates and different types of polyurea [25]. It is now well established by various studies that the polyurea layer can provide considerable protection for metal and concrete substrates [26,27,29,35,36]. Despite the encouraging results achieved with metal substrates, relatively few studies have explored how polyurea performs when it is applied to ceramic armors that are subjected to projectile impact. Moreover, the effect and mechanism of polyurea on the ballistic performance of ceramic composite armor remains unclear. Therefore, it was envisioned that the ballistic resistance of traditional ceramic/metal armors might be substantially improved by coating them with a polyurea layer, to constitute a new armor system without adding unnecessary weight. Consequently, it is necessary to further improve the ballistic performance of ceramic armors coated with polyurea when subjected to projectile impact. Due to the complexity of the behavior of ceramic composite armors under a ballistic impact, it is difficult to establish an analytical model of composite armor containing multi-layers. The existing numerical models are challenging to predict the dynamic failure characteristics of ceramic and polyurea accurately. Hence, it is necessary to explore the ballistic performance of this novel armor system containing multi-layers by using an experimental methodology.

The advanced concept of designing a novel armor system was proposed to address the above issues. For the present study, ballistic experiments were conducted on polyurea-coated ceramic armor that is subjected to projectile impact, to evaluate the effect of the polyurea layer. High-speed photography was used to record the process of the projectile passing through the target. As part of this study, the microstructure evolution of polyurea and the failure patterns of the targets are presented. The results can help in the structural optimization of novel ceramic composite armor. They can also facilitate an understanding of the protective mechanism of ceramic armors.

## 2. Experimental Materials and Target Configuration

### 2.1. Material Characteristics of the Testing Target

The properties of the material used in the experiment are shown in Table 1, and its chemical composition is shown in Table 2. Polyurea is an elastomer synthesized by the rapid reaction of isocyanate and amine polyether at a volume ratio of 1:1, with the microphase structure separated into soft and hard segments. Due to its excellent performance regarding energy dissipation, polyurea is widely used for mitigating ballistic impacts. Specifically, the polyurea used in the present study has an elongation of 400%, with a density of 1125 kg/m^3^. The mixture component of polyurea was sprayed on the substrates using the reaction spraying machine. The armor ceramic was made from silicon carbide ceramic, using high-purity submicron silicon carbide by spray granulation via compression molding, followed by pressureless sintering. The purity of the silicon carbide raw material of ceramics was more than 99.00%. Armor steel, which is widely used in military equipment, was chosen for the metal backplate. The witness targets were made of aluminum 6061-T6 and shaped into a block with dimensions of 150 mm, 150 mm, and 250 mm. Note that the resistance mechanism of the ceramic armor against impact was greatly influenced by the dimensions and material properties of projectiles. The projectile was made of tungsten alloy formed into the shape of a cylinder, which was 8 mm in diameter and 30 mm in length.

### 2.2. Fabrication Process of Testing Target

*Preparation:* Following the process requirements, the necessary raw materials, including silicon carbide ceramics, armor steel plate, and two-component polyurethane adhesive, were prepared. The silicon carbide ceramics was first wiped with a cleaning agent. After the cleaning agent dried, the silicon carbide ceramics were wiped with the special primer and then set aside to dry. The armor steel was polished and cleaned to remove the surface rust. Then, it was wiped with a special primer of polyurethane adhesive.

*The composite phase:* The polyurethane adhesive was evenly spread on the armor steel plate, and the silicon carbide ceramics were bonded to the armor steel, following the arrangement shown in the schematic. The bonded composite plates were placed in the vacuum bag, then the air in the bag was extracted by a vacuum pump to ensure the composite compactness of the armor steel plate with the armor ceramic. After the composite plate was completely dry, it was removed from the vacuum bag, and the spilled glue on the side of the composite board was cleaned up with a blade.

*Spraying phase:* The glue stains on the surface of the composite plate were cleaned, and the surface of the armor steel was wiped with the steel plate primer specially designed for polyurea. After the primer was dried entirely, polyurea was sprayed on the armor steel/ceramic. After the polyurea coating was dried, the thickness of the selected point was measured. The polyurea coating was smoothed over with an air mill to ensure consistency of the coating thickness. The above steps were repeated until the requirements of coating thickness were met. The compliant composite board was placed in the drying room until it was completely dry.

### 2.3. Target Configuration

Four kinds of composite ceramic armor were designed with different thicknesses and layer positions of the polyurea layer, as shown in Table 3 and Figure 1. Note that seven reference tests were conducted. Among them, the lowercase p, c, and s stand for polyurea, ceramic, and armor steel, respectively. The number indicates the thickness of the material, and the unit is in millimeters. The capital letters F, M, and B indicate that the polyurea layer is located in the front, middle, and rear sides of the target, respectively. The capital letter C denotes the ceramic/metal armor without a polyurea layer.

## 3. Experimental Setup and Testing Method

Figure 2 shows the experiment that was performed at the State Key Laboratory of Explosion Science and Technology, Beijing Institute of Technology, China. This experimental system was based on a ballistic gun fitted with velocity-testing and photoelectricity-testing systems. The witness target was fixed to an immobilized platform, using rigid clamps at the edges, at a distance of 3.5 m from the muzzle. The projectiles, formed as a cylinder of 8 mm in diameter and 30 mm in length, were launched at a velocity range of 700–950 m/s by adjusting the mass of propellants used in each cartridge. Using the fitted high-speed camera, each penetration process of the projectile was recorded.

The experimental procedure was separated into two parts: (1) establishing the relationship between the reference depth of penetration and the velocity of the projectile, and (2) testing the ballistic performance between different target configurations within the velocity range. The module test was conducted to investigate the effectiveness of the polyurea layer and the influence of its coating position. Instead of testing the residual velocity, reference tests were conducted because the debris of the ceramics generated from the penetration process may have interfered with the tinfoil paper. A series of reference tests were conducted to establish the penetration depth in the witness targets and to evaluate the ballistic performance of different configurations of targets. The witness targets were fixed to the frame and treated as half-infinity targets of adequate dimensions. Figure 3 shows that the projectile penetrated the witness target after passing through the tested target, to ensure the independence of each penetration process. The purpose of the reference test is to establish the penetration depth of the projectile in the witness target at different velocities, and to measure the ballistic performance of the polyurea-reinforced ceramic/metal armor, as shown in Figure 3, where $P_{r}$ represents the residual penetration depth into the witness block in the module test, and $P_{\mathrm{wit}}$ is the penetration depth into the witness block in the reference test.

It should be noted that twenty-seven projectiles were launched in this study, including seven shots for reference tests and twenty shots for module tests. Five shots were carried out for each configuration of the targets. After the test, the targets were cut along the central line using a wire cutter, and the penetration depth was obtained using the Vernier scale. The protective effect of the tested targets was assessed, based on mass efficiency, which is calculated as:(1)$$N_{E}=\frac{\left(P_{\mathrm{wit}}-P_{r}\right)\cdot \rho_{\mathrm{wit}}}{AD_{\mathrm{target}}}$$
where $AD_{\mathrm{target}}$ denotes the areal density of the testing target and $\rho_{\mathrm{wit}}$ is the density of the witness target. In the formula, both $P_{\mathrm{wit}}$ and $P_{r}$ are the penetration depth of the projectile at the same velocity. Due to its being difficult to accurately control the projectile’s speed in the experiment, to achieve the same speed in both of the two rounds, the linear fitting value of $P_{\mathrm{wit}}$ is used to calculate the mass efficiency $N_{E}$.

## 4. Experimental Results

The penetration depth on the module test and reference test are shown in Figure 4a,b, respectively. A linear best-fitted analysis was conducted to correlate the projectile velocity and penetration depth in the witness target. As shown in Figure 4a, the penetration depth in the witness targets of each configuration was obtained. It should be noted that attitude deflection of the projectiles occurred on two shots in the case of configuration C.

There was a noticeable difference in the penetration depth among each configuration. Figure 4a clearly shows that the DOP of configuration C was slightly deeper than that of the other three groups for the velocity range of 700–825 m/s. This implies that the bulletproof performance of polyurea-reinforced ceramic armor is not weaker than that of ceramic/metal armor. It is noteworthy that configuration F showed better protective performance, due to the relatively small penetration depth. Considering the weight advantage of the polyurea-reinforced ceramic armor, it can be seen in Figure 4c that the mass efficiency of configuration C was lower than in the other three groups. Overall, these results demonstrated that when the velocity range of the projectile was 700–825 m/s, the ballistic performance of polyurea-reinforced ceramic armor was slightly better than in the ceramic/metal armor. The mass efficiency of the composite armor in configuration F was improved from 6.69 to 12.68, compared with that in configuration C, which increased by 89%. Thus, adding polyurea was promising in terms of the performance improvement of the novel composite armor.

Figure 5a shows that the perforations of the tested targets varied among the different configurations. Note that the horizontal line in Figure 5a is the projectile diameter. There was a significant difference in the perforation diameter of the polyurea layer among each group. The perforation size of the polyurea in configuration F was approximately equal to the projectile diameter. It is noteworthy that the perforation size of the polyurea in configuration M was nearly equal to zero, which means that there is a self-healing failure mode induced by the rubbery behavior of the polyurea. As shown in Figure 5b,c, the ceramic layer was completely fractured and detached from the substrate, and the projectile was shown to have a mushroom-shaped head.

As shown in Table 4, the deformation process of the target during penetration was recorded by a high-speed camera. During the impact of the projectile on the composite armor in configuration F, particles were produced due to the tensile fracture of the ceramics, and the flash was produced by electromagnetic radiation. Table 4 also shows that a large amount of ceramic powder was ejected from the perforation after the projectile passed through the target plate, accompanied by the impact flash. From moment 2 of configuration F, the local reverse deformation response of polyurea indicated its high elasticity behavior. From moment 3, the local deformation of the reverse deformation of the polyurea spread from the central area to the global area. Note that this local deformation was induced by the reverse movement of the ceramic fragments in the central area. In addition, the maximum of the global reverse deformation was achieved at moment 3. At moment 4, the partially reversed movement of the ceramic powder through the polyurea perforation due to the rebound of the armor steel can be observed. Then, at moment 5, a large amount of ceramic powder can be seen diffusing from the edge of the target, indicating that the edge area of the ceramic layer is almost completely compromised at this moment. Thus, the powder in the central region of the ceramic can spread outward through the edge from the central region of the impact of the projectile target. During this process, the polyurea layer exhibits a shear failure and its pore diameter is the same as that of the projectile, as shown in Figure 5a. It can be inferred from the limited deformation of the polyurea that the glass transition stage was induced by the projectile’s directly impacting it [37]. Thus, the glass transition and the confinement behavior of polyurea on ceramic can explain the superior protection performance of configuration F.

The intensity of the flash phenomenon in configuration M was obviously lower than that in the other three groups, as shown in Table 4. This reason could be that the polyurea elastomer appeared on the back of the ceramic layer. Thus, it was unable to provide support to the ceramic as effective as that of armor steel, which reduces the pressure between the projectile and the target and mitigates the interaction intensity. Thus, the flash produced by electromagnetic radiation during the impact process is reduced. At moment 2, it can be seen that the ceramic powder splashed along the reverse direction of the projectile. Due to the rebounding of the backplate, the ceramic powder bounced from the central region to the edge of the plate. Note that, from moments 3 to 5, the ceramic particles gradually increased. The composite target in Configuration B was passed through by the projectile at moment 2, and the back polyurea layer reached maximum deformation at moment 3. The deformation characteristics showed that the back polyurea layer was separated from the armor steel because the armor steel could not achieve a deformation as great as that of the polyurea. In Configuration C, the flash produced during the contact process between the projectile body and the ceramic was the strongest because the ceramic layer was the thickest among the groups. In addition, a large quantity of ceramic powder can be observed in Configuration C. The ceramic fragments moved in the opposite direction from the center to the edge, similar to Configuration M, accompanied by a small amount of deformation and the substantial rebound of the armor steel at the back.

Table 5 shows that the failure morphologies were concentrated around the perforation, which indicated that the glass transition state of polyurea was induced during the impact. The macroscopic failure morphology of the composite targets is provided in Table 5 to analyze the terminal failure characteristics. The failure morphology of the target plates varied among the different configurations. It was noted that the large-area collapse of the ceramics occurred under the impact of the projectile. A compressive stress wave was produced during the contact process of the projectile and target. After the projectile impacted the composite plate, a compression stress wave was launched into the target plate along the direction of the projectile’s motion. When the compression wave reached the interfaces of the materials, the compressive stress wave was incompletely reflected in the opposite direction. As the wave was reflected, a corresponding tensile wave formed in the ceramic, causing its fragmentation, and the crack extended to the edge of the target plate. As a result, a large area of ceramic collapse was formed. In the target plates of configurations B and C, after the projectile body impact, all the ceramics in the faceplates collapsed. In configuration F, the ceramics appeared in the middle interlayer and all the ceramics collapsed. In configuration M, some ceramics were still retained in the armor steel and the polyurea layer, which formed an imperfectly collapsed pattern.

## 5. Failure Mechanisms

### 5.1. Perforation of Targets

The failure modes of different target plates were analyzed to study the influence of polyurea coating position on the failure mode. The results are provided in Table 6. In configuration F, the perforation of the polyurea layer was round and its diameter was similar to that of the projectile, as shown in Figure 5a. On the polyurea layer, there was no apparent damage around the perforation. In configuration F, the morphologies of polyurea perforation under the impact of the different projectile velocities were similar. The failure feature of configuration F is that the polyurea layer showed local shear failure under the impact of the projectile body with the glass transition. The perforation of the armor steel back was also petal-shaped. It is worth noting that in configuration M, the polyurea layer appeared to rebound after stretching, which is characteristic of self-healing. During the penetration process, the polyurea exhibited high elasticity behavior with substantial deformation. When the projectile passed through, the perforation shrunk, which showed as a tiny hole, and the appearance of the surface was rough within a distance three times that of the diameter around the perforation. In configuration B, the perforation diameter of the polyurea layer was slightly smaller than that of the armor steel, which showed that, after stretching, both the polyurea layer and the armor steel experienced the rebound phenomenon to varying degrees. Nevertheless, the pore diameter of the polyurea layer was much larger than that in configuration M. Note that configuration B is different from the other three groups. In addition, in configuration B, three different failure modes exist: (1) the polyurea layer has an everted perforation in cross-section, in which the edge of perforation is irregular. (2) The failure mode of B2 is based on B1, and the edge cracks continue to propagate outward with the range of propagation, reaching 3–5 times the projectile diameter and radiating around the perforation. (3) B5 only exhibited a smooth local perforation, accompanied by local uplift. The area around the bullet hole is also relatively tidy, with no eversion of the edge.

### 5.2. Damage Pattern of the Targets

Based on the four types of projectile penetration into the targets, five damage modes were proposed, as shown in Figure 6. The five damage modes were shearing-hole, self-healing, spallation, perforation, and cracking.

Damage pattern a: localized shearing failure appeared, with an approximately equal diameter of polyurea layer perforation and equal projectile diameter, with no apparent tensile or compressive damage around the round hole. Note that the appearance implies the glass transition of polyurea.

Damage pattern b: the self-healing failure mode exhibited a tiny hole and rough morphology around the polyurea layer. This morphology was formed when the projectile passed through the target, and the polyurea rebounded after a large stretching deformation.

Damage pattern c: the spallation failure mode was demonstrated by the debonding between polyurea and armor steel, with a tension shear fracture and a smooth edge and irregular shape.

Damage pattern d: the perforation failure mode exhibited an eversion at the fracture of the polyurea layer.

Damage pattern e: this damage pattern was similar to damage pattern d, with the extension of the cracks.

Scanning electron microscopy (SEM), the method used to observe the microstructure of the sample surface, was employed to analyze the failure mechanisms of polyurea and its fracture morphology. To establish the damage mode of the polyurea coating, SEM was used to scan the samples at the point of polyurea fracture under different positions in the layers and penetration velocities. The damage mode of the polyurea coating and its effect on ballistic performance was analyzed in combination with the ballistic test results. To ensure the scanning position of the specimen, a cube with a 1 cm side length was cut out with a water jet, then the fracture area was reserved for SEM scanning. As shown in Table 6, the cutting position was inside the yellow box, ensuring that the sample surface was clean and dry. Then, a gold layer was sprayed onto the polyurea surface. Multiple positions were observed on each fracture surface at different magnifications. As shown in Figure 5a, the perforation diameter and the shape of the polyurea layer in configuration F were similar to those of the projectile. This similarity can be explained by the glass transition induced by local shear failure at a high strain rate. At low magnification, it is clear that the fracture surface contains a large number of microvoids of different sizes, as shown in Figure 7. The initial state of the polyurea is shown in Figure 7b. At high magnification, it can be seen that there is a large amount of debris distributed around the hole.

As shown in Figure 8a,b, incomplete stretching and deformation holes in configuration M were demonstrated. As shown in Figure 8c,d, stepped cracks in the polyurea and ceramic debris embedded in the polyurea appeared, which were formed during the tearing process of polyurea caused by the combination of compression and tensile waves. Debris was distributed in deformation holes and lamellar structures, as shown in Figure 9a,b. The propagation cracks in different directions were distributed around the hole, as shown in Figure 9c.

## 6. Conclusions

In this study, a polyurea-coated ceramic armor was subjected to projectile impact and investigated to evaluate the ballistic performance of different configurations of the targets. The depth of penetration and the mass efficiency of the tested armor were quantified. Subsequently, the failure mechanisms of polyurea-reinforced ceramic armor and ceramic/metal armor were detailed for different configurations, and the deformation processes of the targets were analyzed. Based on the experimental results, five patterns of targets were also summarized. The following conclusions can be drawn from the results:(1)The polyurea layer shown could improve the ballistic performance of ceramic composite armor. The mass efficiency of polyurea-coated ceramic armor was found to be higher than that of ceramic/metal armor. The position of the polyurea layer also has a significant influence on ballistic performance. The targets in configuration F showed better ballistic performance. Compared with ceramic/metal armor, the mass efficiency of composite armor with a polyurea layer on the front face was improved from 6.69 to 12.68 on average at the same speed range (700–825 m/s), and increased by 89%.(2)The damage characteristics of the different configuration targets are presented. It was found that the damage characteristics of the polyurea layer are related to its position in the layers, which influenced the failure mode of the targets.(3)Five failure patterns are presented from the experimental conditions: the shearing-hole, self-healing, spallation, perforation, and cracking.(4)SEM analysis of the polyurea layer was conducted. The results of the microscopic analysis show that the polyurea in configuration F demonstrated the micro-morphology of glass transition, which explains the high ballistic performance of targets in configuration F.

## Figures and Tables

**Figure 1 materials-15-03918-f001:**
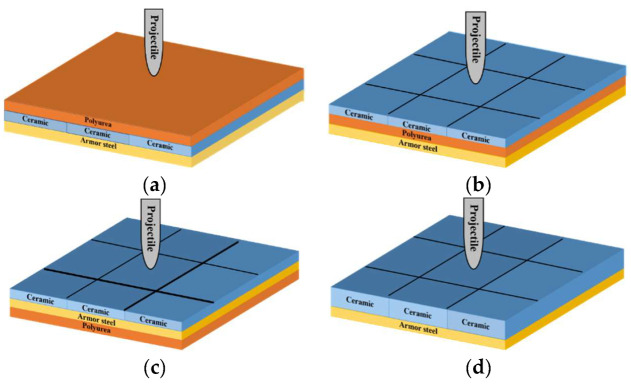
Schematic of the targets: (**a**) Configuration F. (**b**) Configuration M. (**c**) Configuration B. (**d**) Configuration C.

**Figure 2 materials-15-03918-f002:**
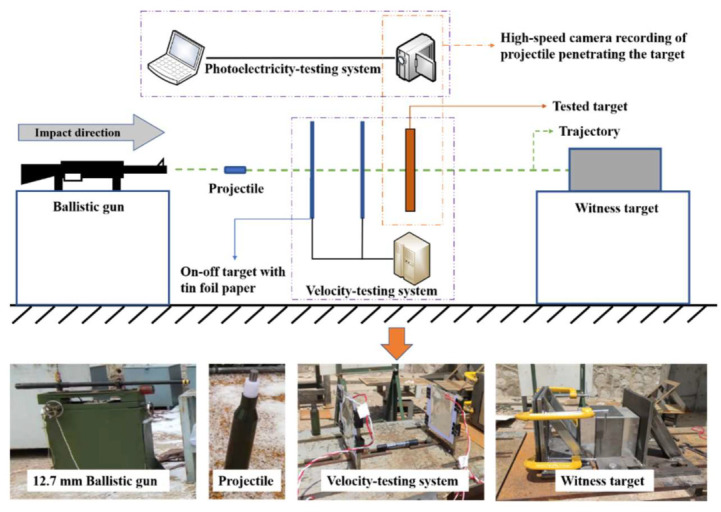
Experimental device layout.

**Figure 3 materials-15-03918-f003:**
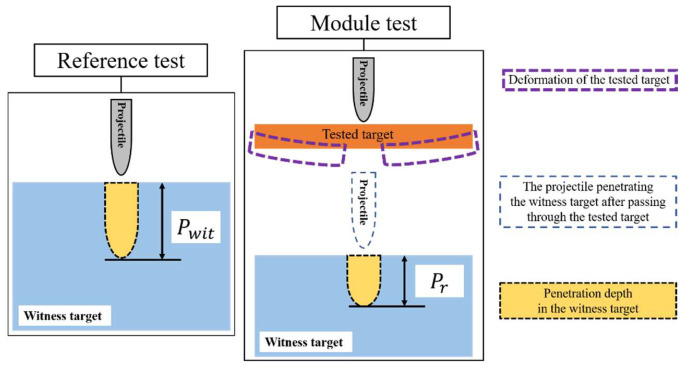
Schematic of the test process.

**Figure 4 materials-15-03918-f004:**
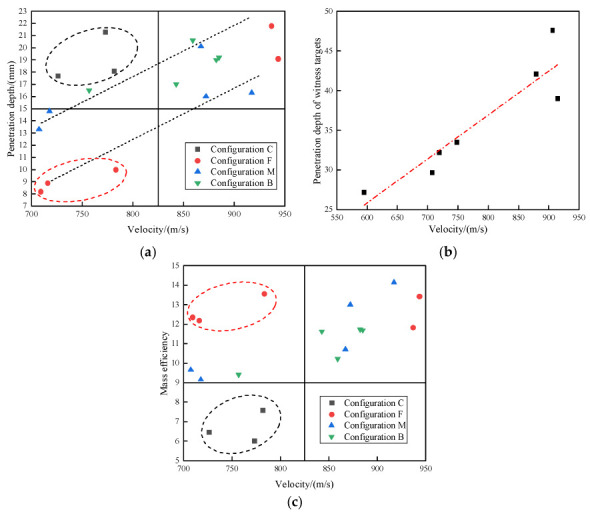
Penetration depth of the different targets. (**a**) Penetration depth. (**b**) The penetration depth of the reference test. (**c**) Effectiveness of the targets.

**Figure 5 materials-15-03918-f005:**
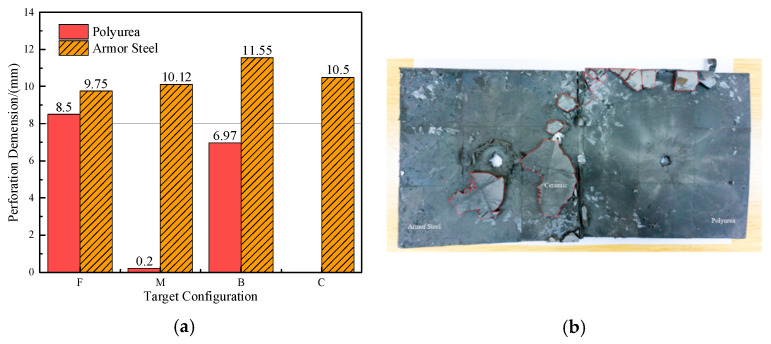
Perforation and morphology. (**a**) Perforation of the target. (**b**) Morphology of the F target. (**c**) Morphology of the projectile and the witness target.

**Figure 6 materials-15-03918-f006:**
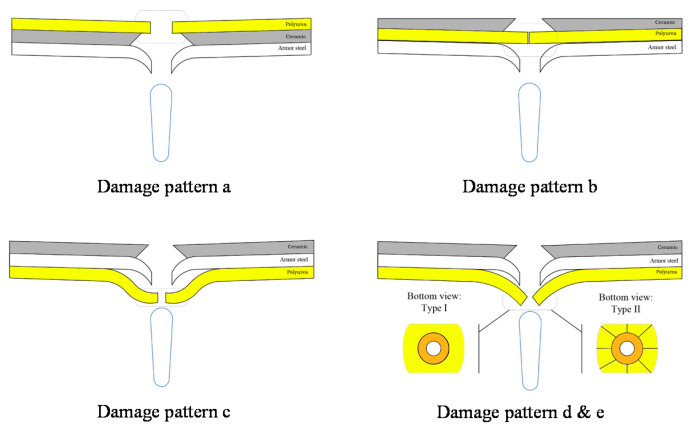
Damage patterns of the targets.

**Figure 7 materials-15-03918-f007:**
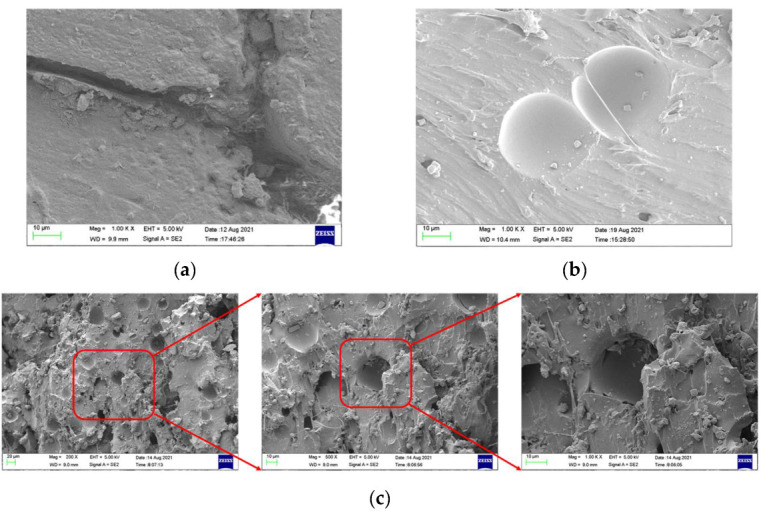
SEM images of polyurea on the target of configuration F and in the initial state. (**a**) Surface area. (**b**) The initial state of the polyurea. (**c**) Fracture area debris.

**Figure 8 materials-15-03918-f008:**
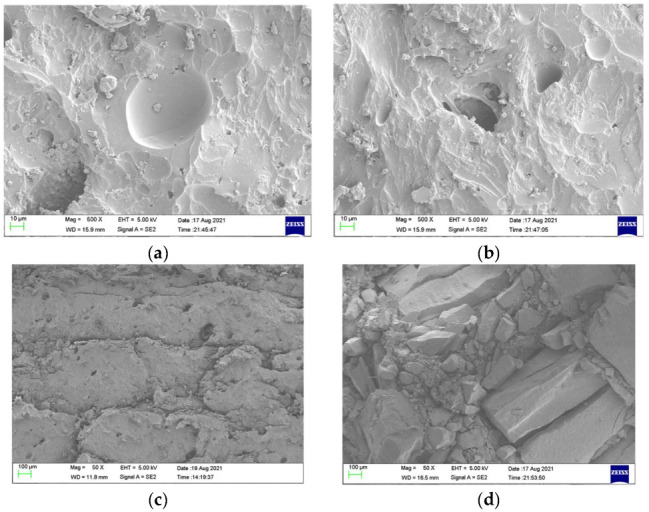
SEM images of polyurea on the target of configuration M. (**a**) Hole. (**b**) Deformation hole. (**c**) Stepped cracks. (**d**) Ceramic debris.

**Figure 9 materials-15-03918-f009:**
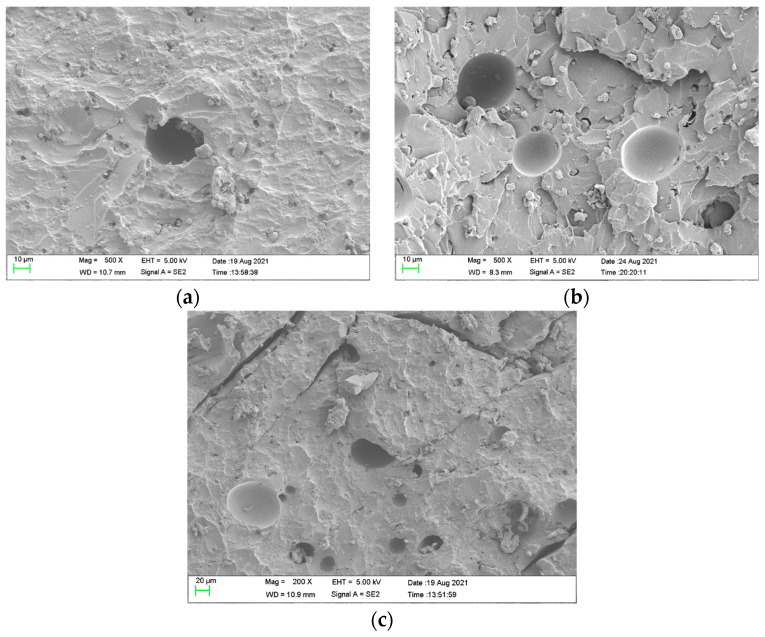
SEM images of polyurea on the target of configuration B1. (**a**) Debris and holes. (**b**) Brittle crack and holes. (**c**) Cracks and holes.

**Table 1 materials-15-03918-t001:** Material properties.

Material	Material Properties					
Polyurea	Density (kg/m^3^)	Solid content	Tensile strength (MPa)	Tearing strength (MPa)	Elongation	Solidification time (s)
	1010	≥96%	10	≥40 KN/m	400%	45
SiC ceramic	Density (kg/m^3^)	Vickers hardness (MPa)	Bending strength (MPa)	Compressive strength (MPa)	Crystal density (μm)	Elasticity modulus (GPa)
	3130	2600	400	2200	5	430
Armor steel	Density (kg/m^3^)	Brinell hardness (HB)	Yeld strength (MPa)	Tensile strength (MPa)	Elongation	
	7850	500	1400	1700	10%	
Aluminum 6061-T6	Density (kg/m^3^)	Tensile strength (MPa)	Yield strength (MPa)	Elongation (%)		
	2850	318	257	9.9		
Tungsten alloy	Density (kg/m^3^)	Yield strength (MPa)	Elongation (%)	Rockwell hardness (HRC)		
	17600	742	8.8	27		

**Table 2 materials-15-03918-t002:** Chemical composition of materials (in wt %).

Material	Chemical Composition								
Armor steel	C	Si	Mn	P	S	Cr	Ni	Mo	B
	0.32	0.4	1.2	0.01	0.003	1.0	1.8	0.7	0.005
Aluminium 6061-T6	SI	Fe	Cu	Mn	Mg	Cr	Zn	Ti	Al
	0.59	0.369	0.246	0.063	1.025	0.201	0.103	0.028	97.37
Tungsten alloy	W	Ni	Fe						
	93	5.1	1.9						

**Table 3 materials-15-03918-t003:** Target design.

Configuration of Targets				
Group	F	M	B	C
Configuration (mm)	(5p)/4.5c/4.5s	4.5c/(5p)/4.5s	4.5c/4.5s/(5p)	10c/4.5s
Areal density (g/cm^2^)	5.464			6.685

**Table 4 materials-15-03918-t004:** High-speed photography of the targets.

Configuration Template	Moment 1	Moment 2	Moment 3	Moment 4	Moment 5
F	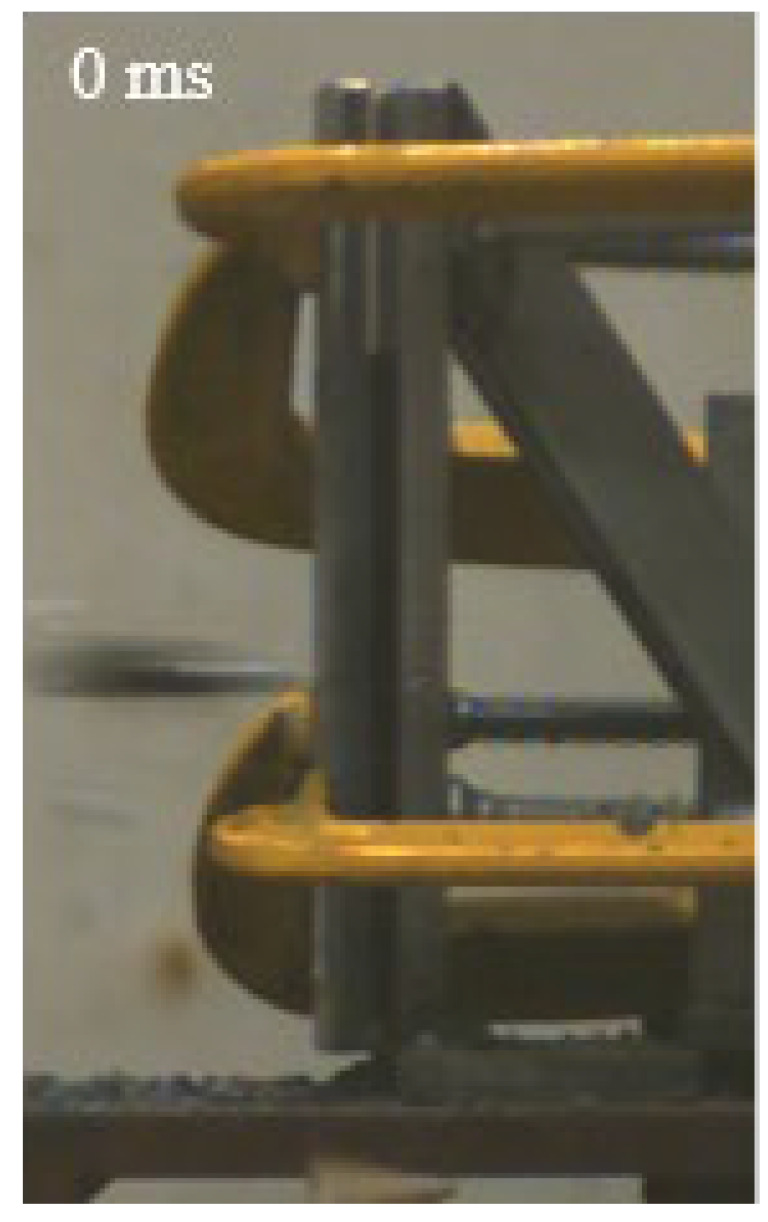	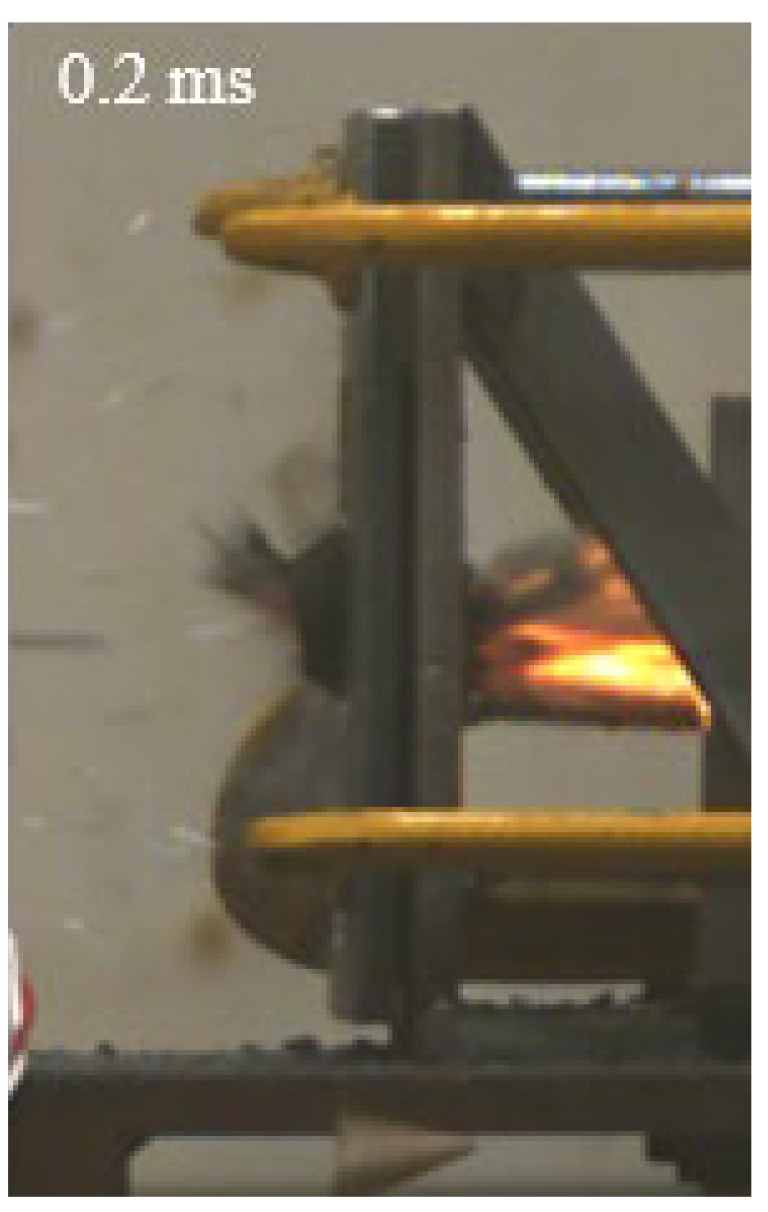	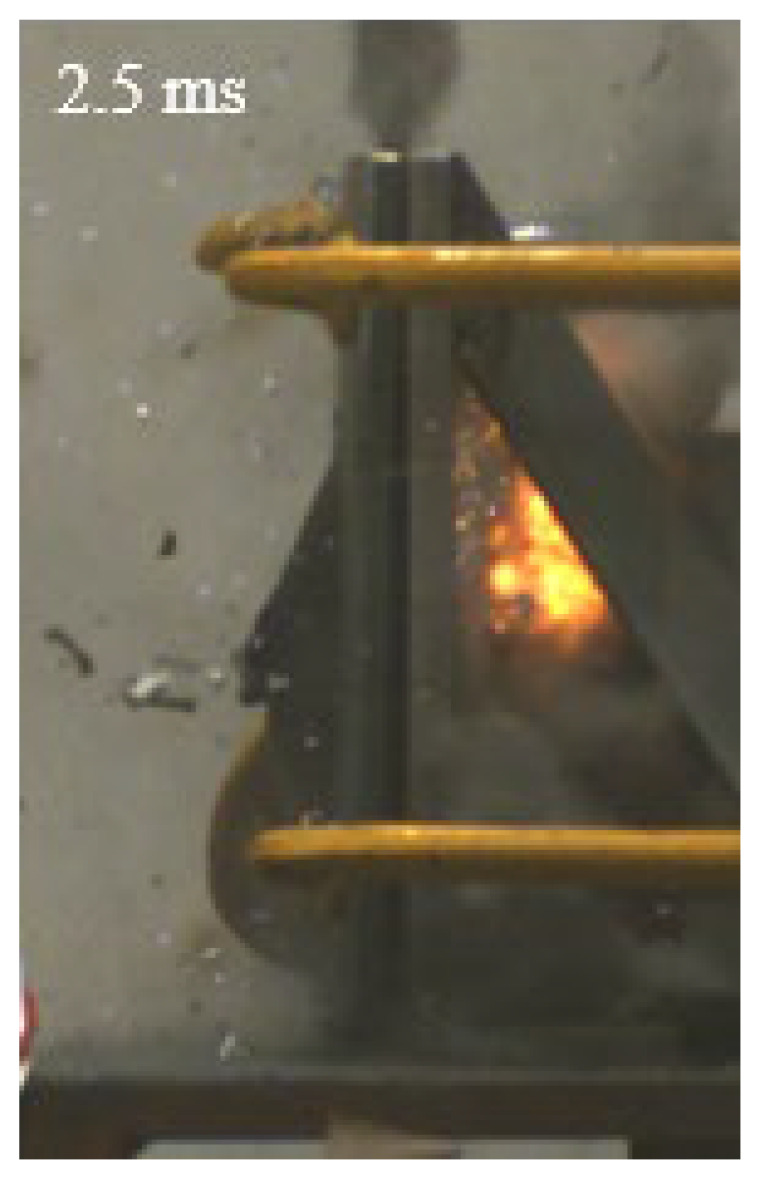	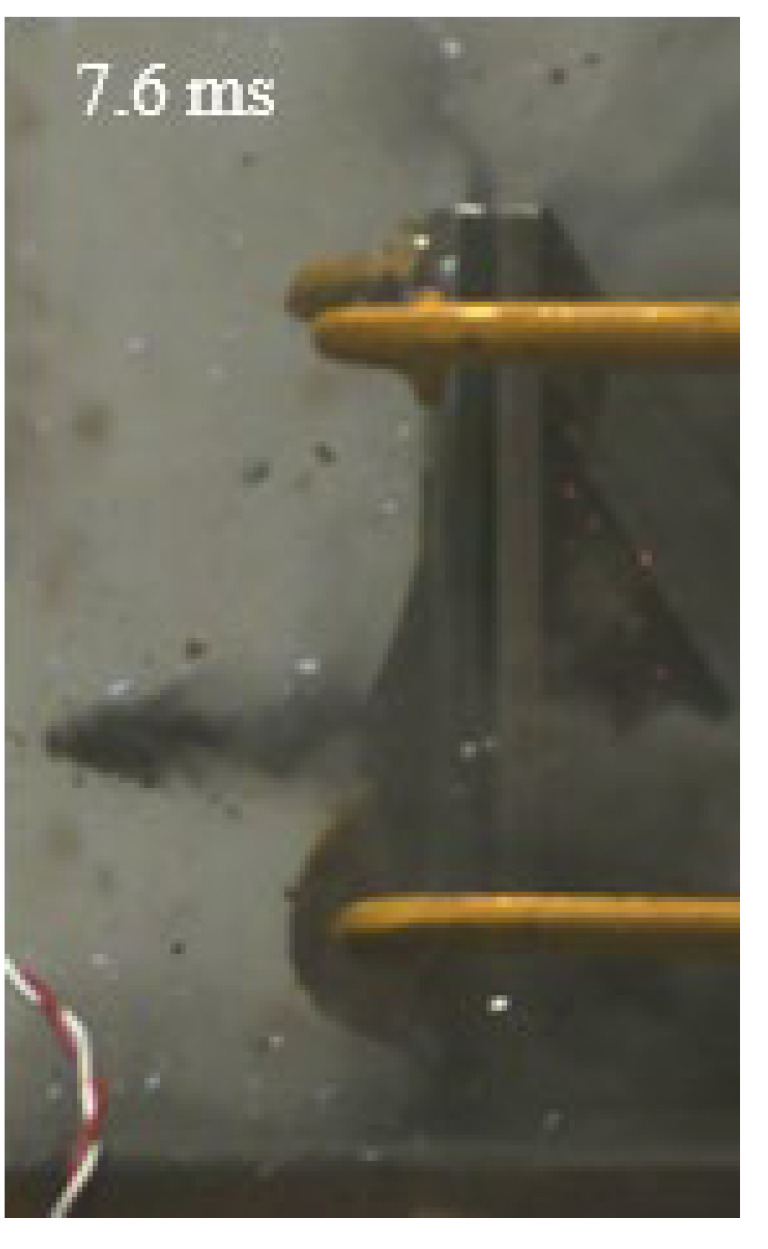	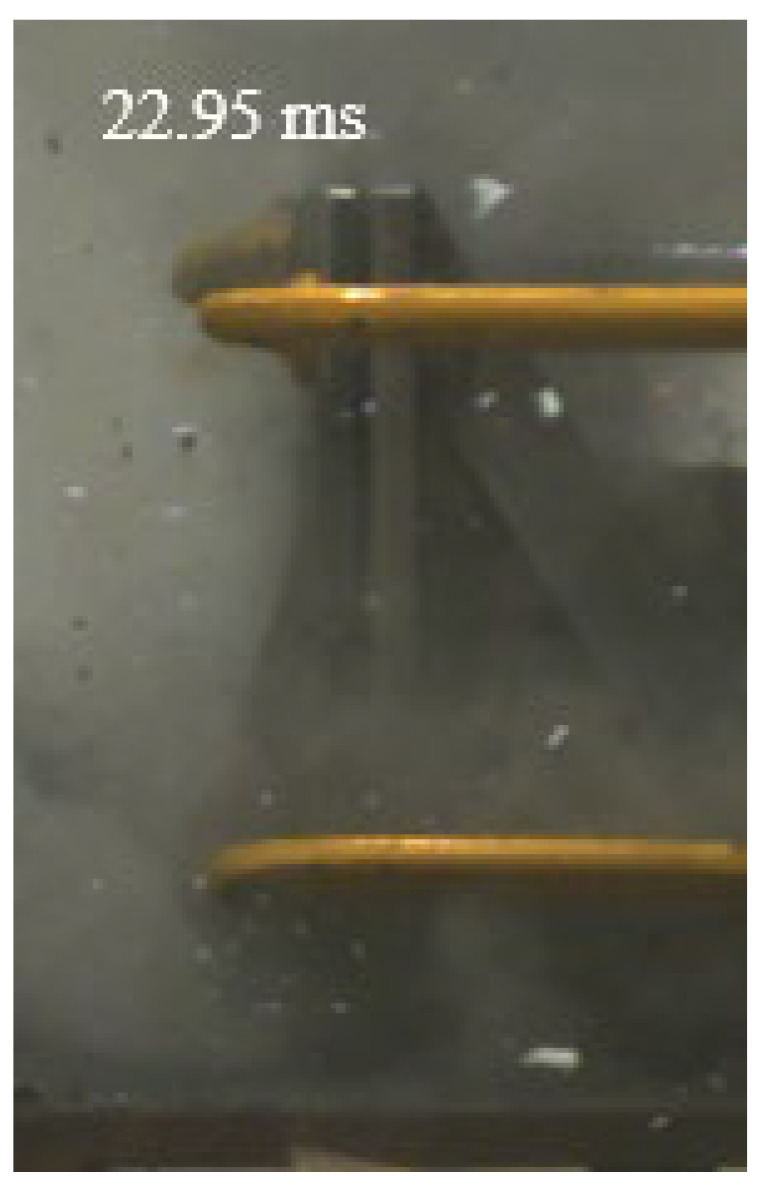
M	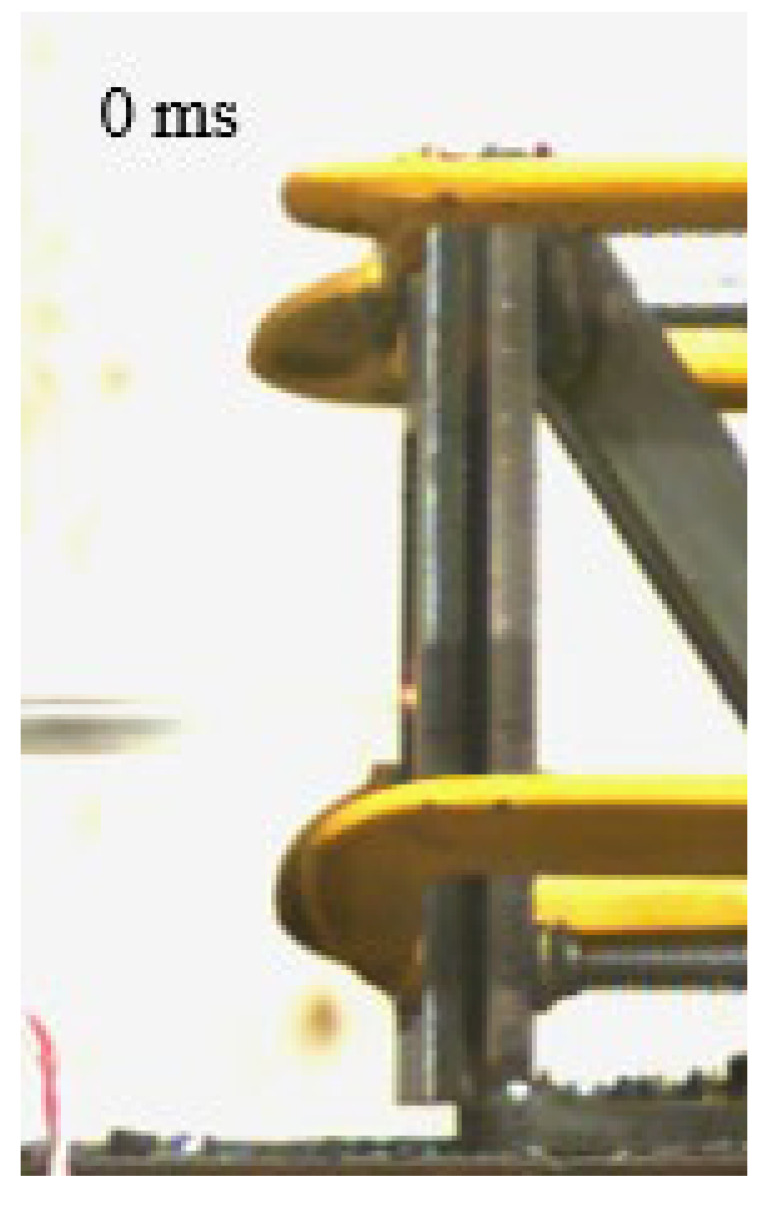	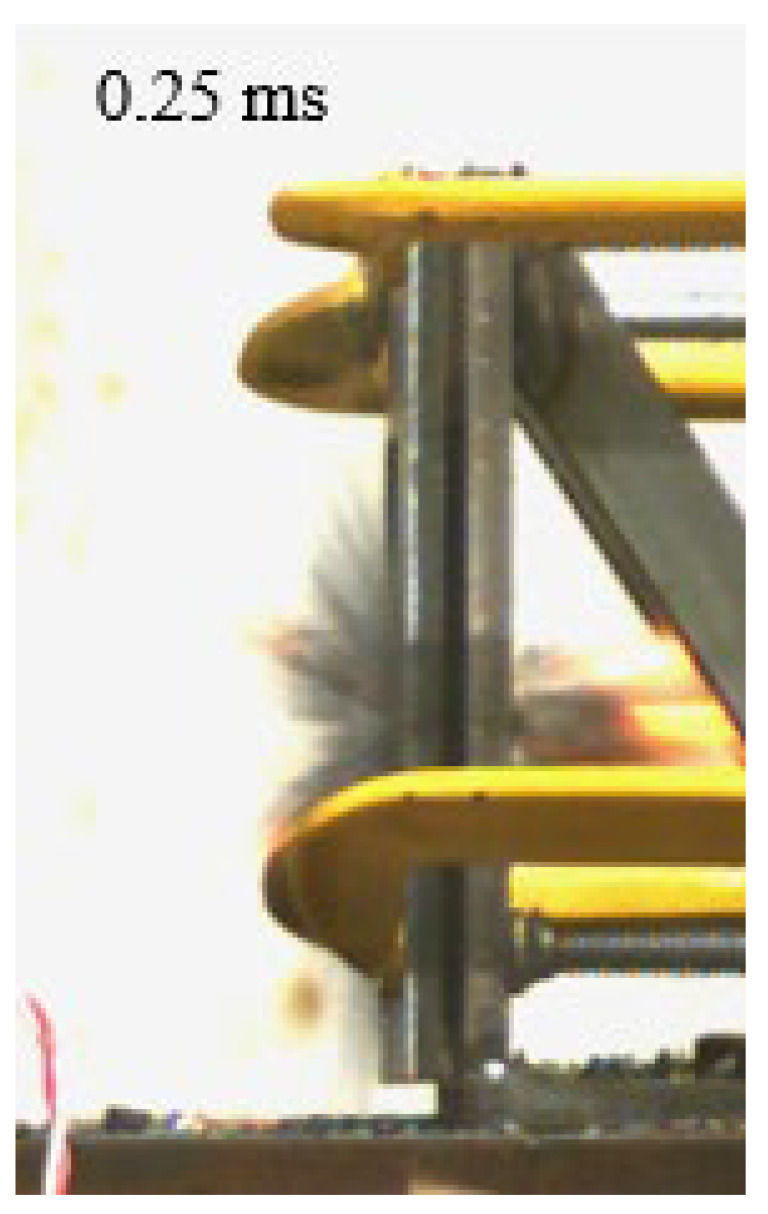	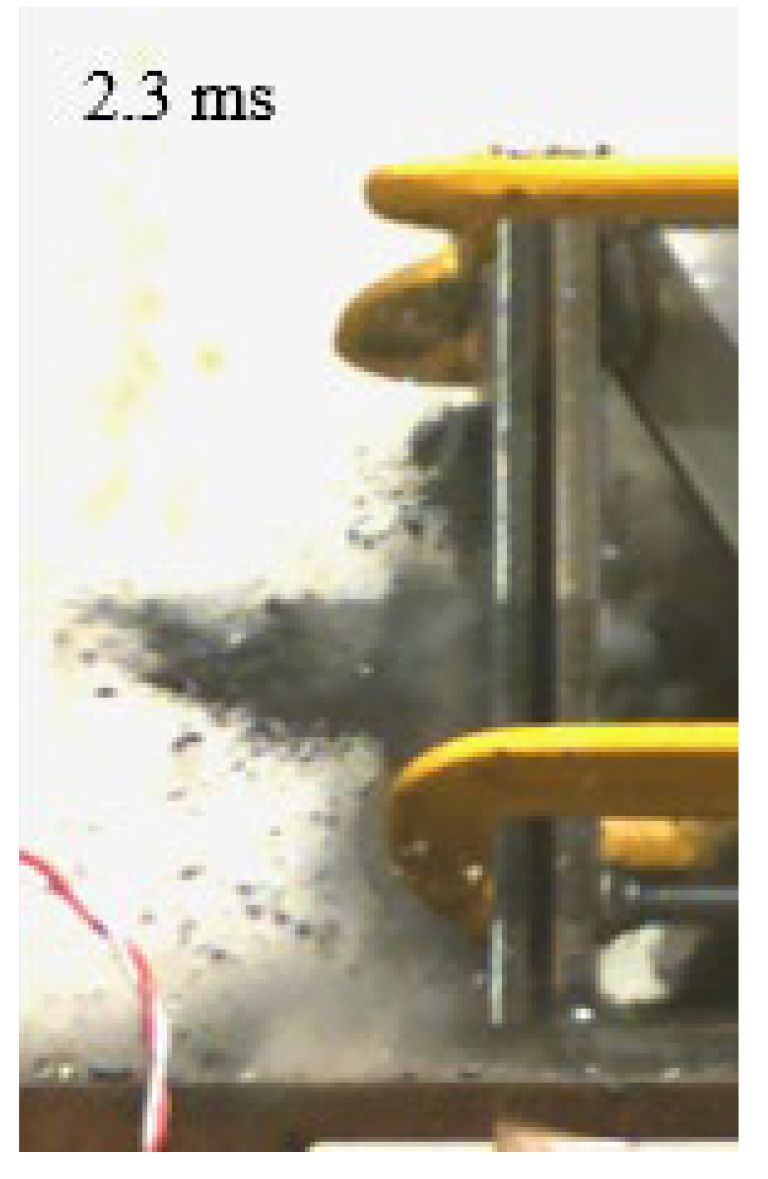	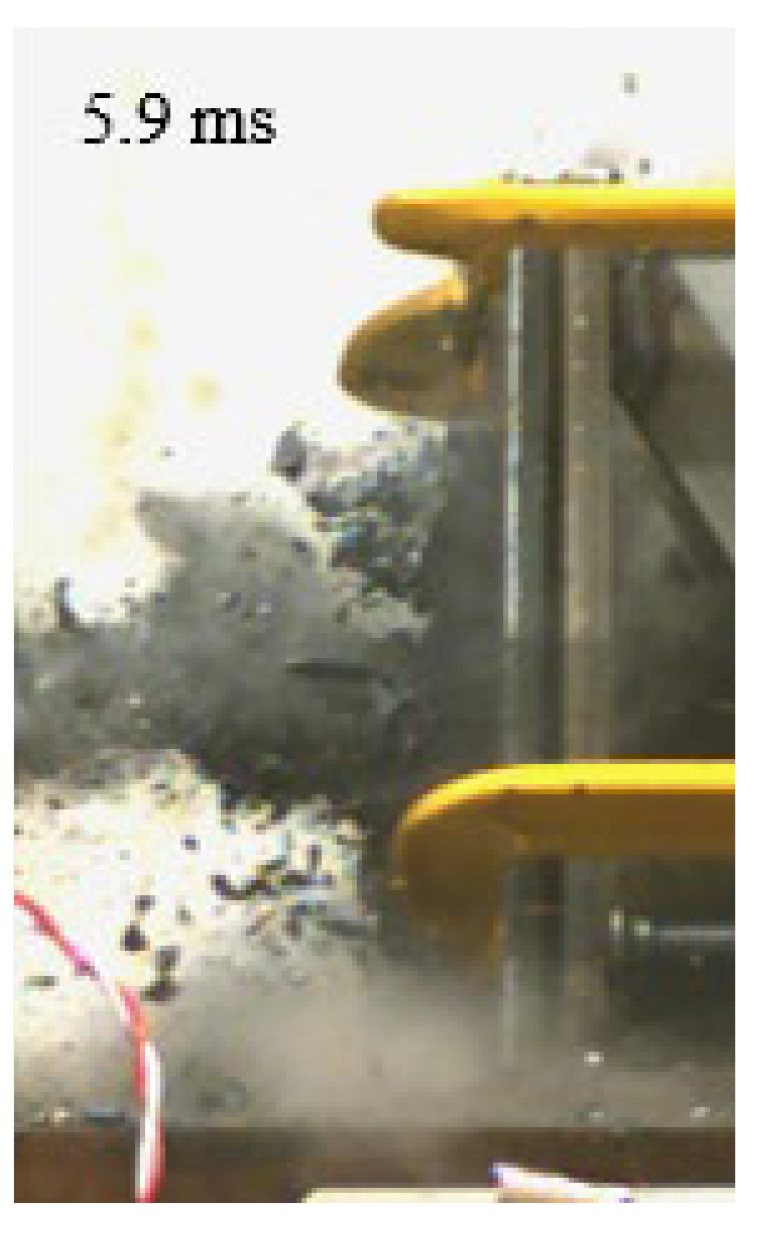	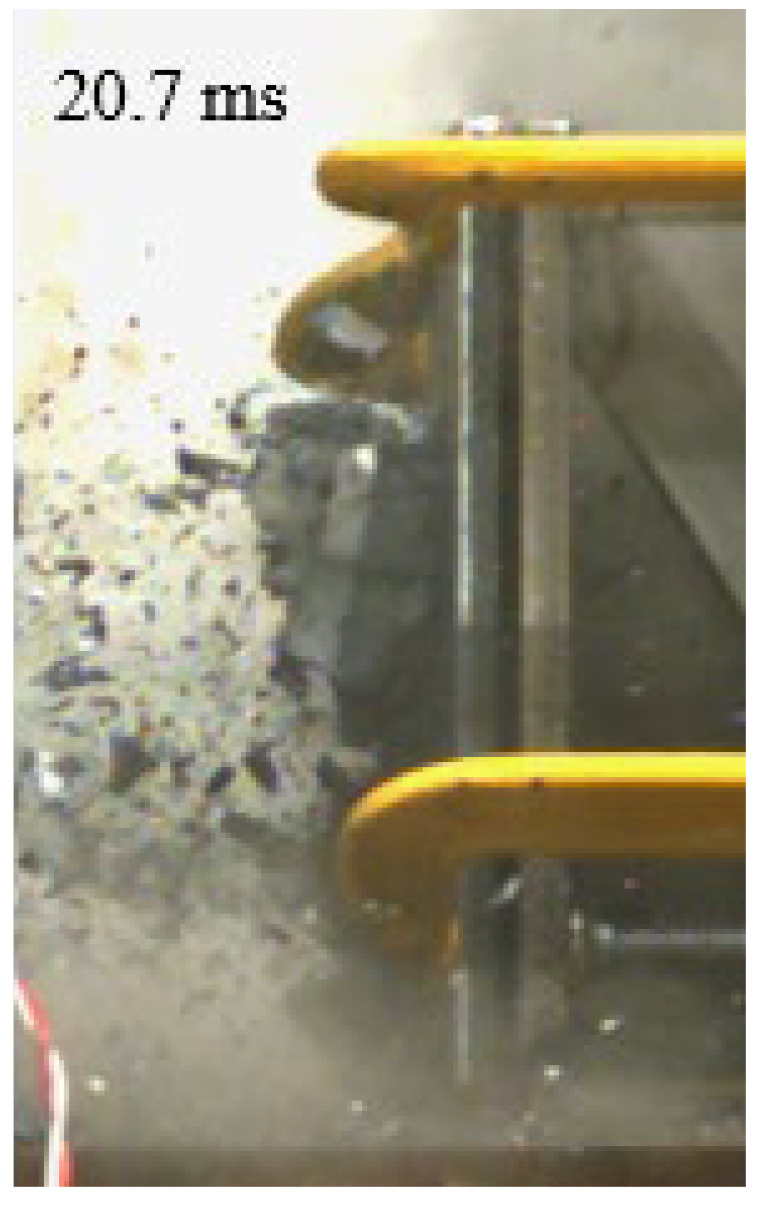
B	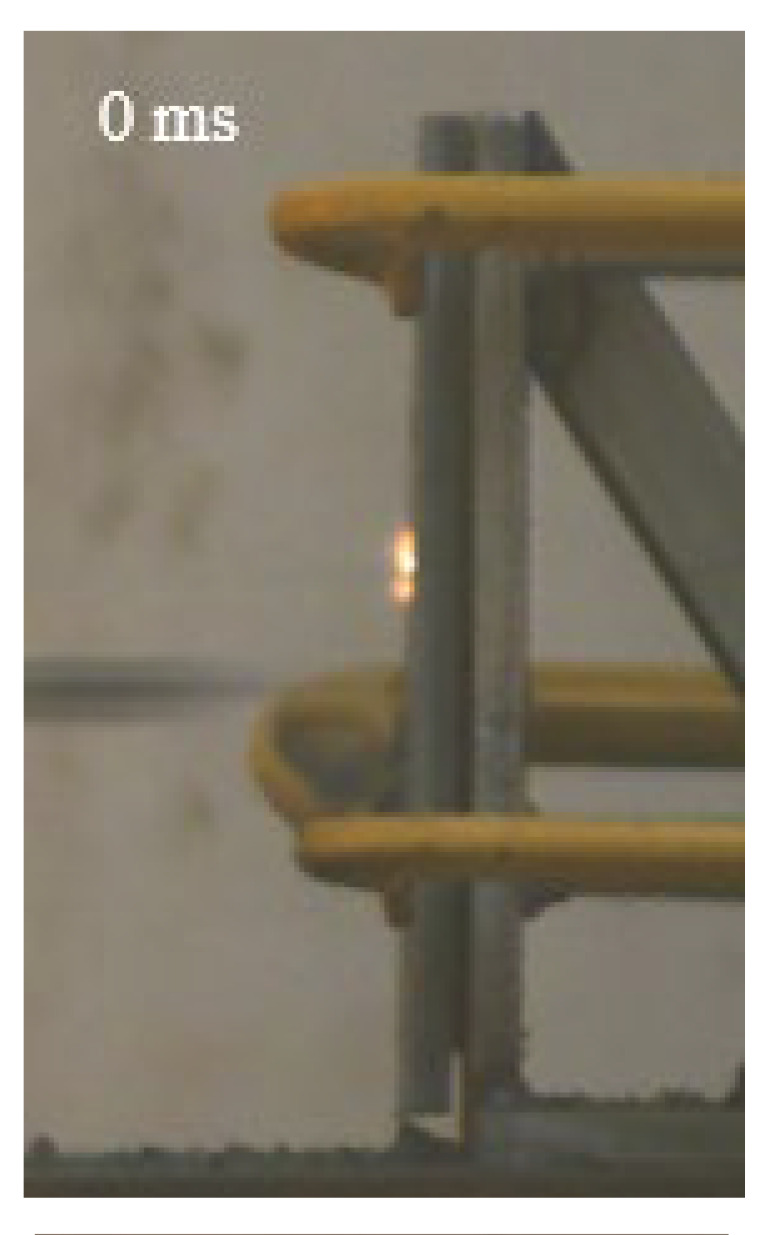	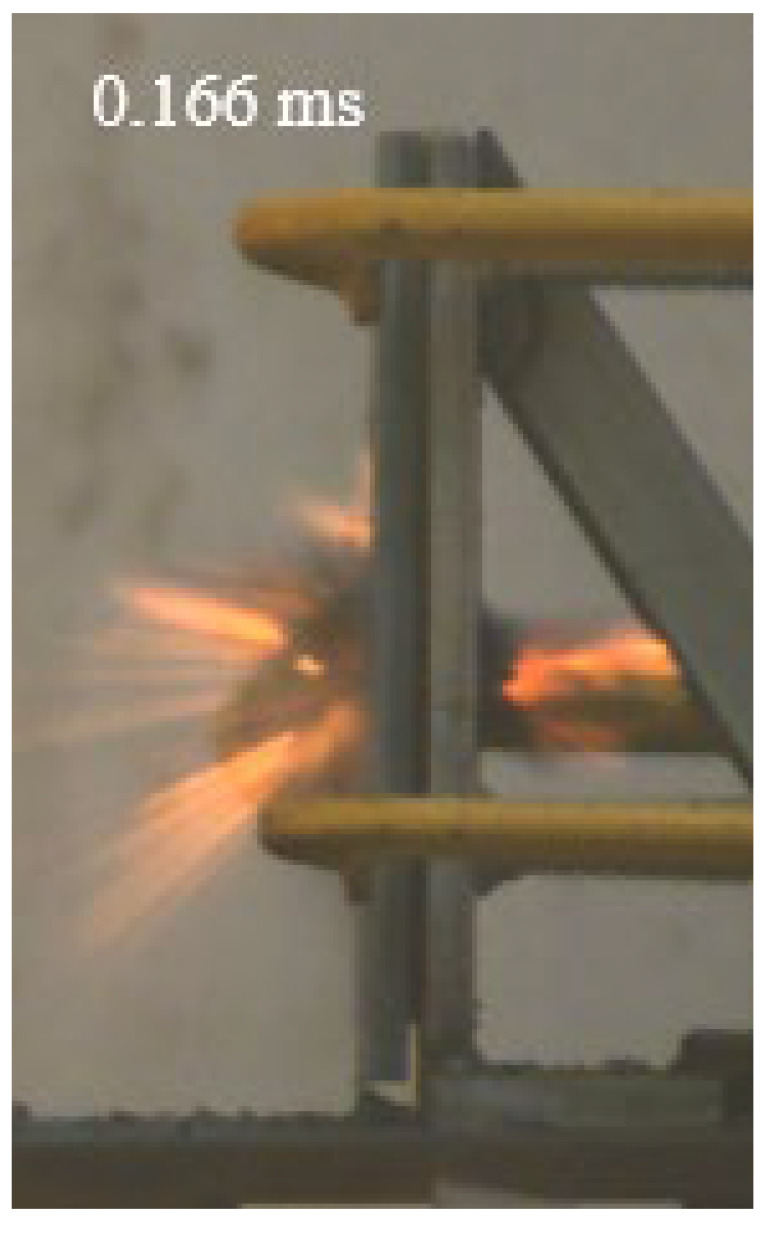	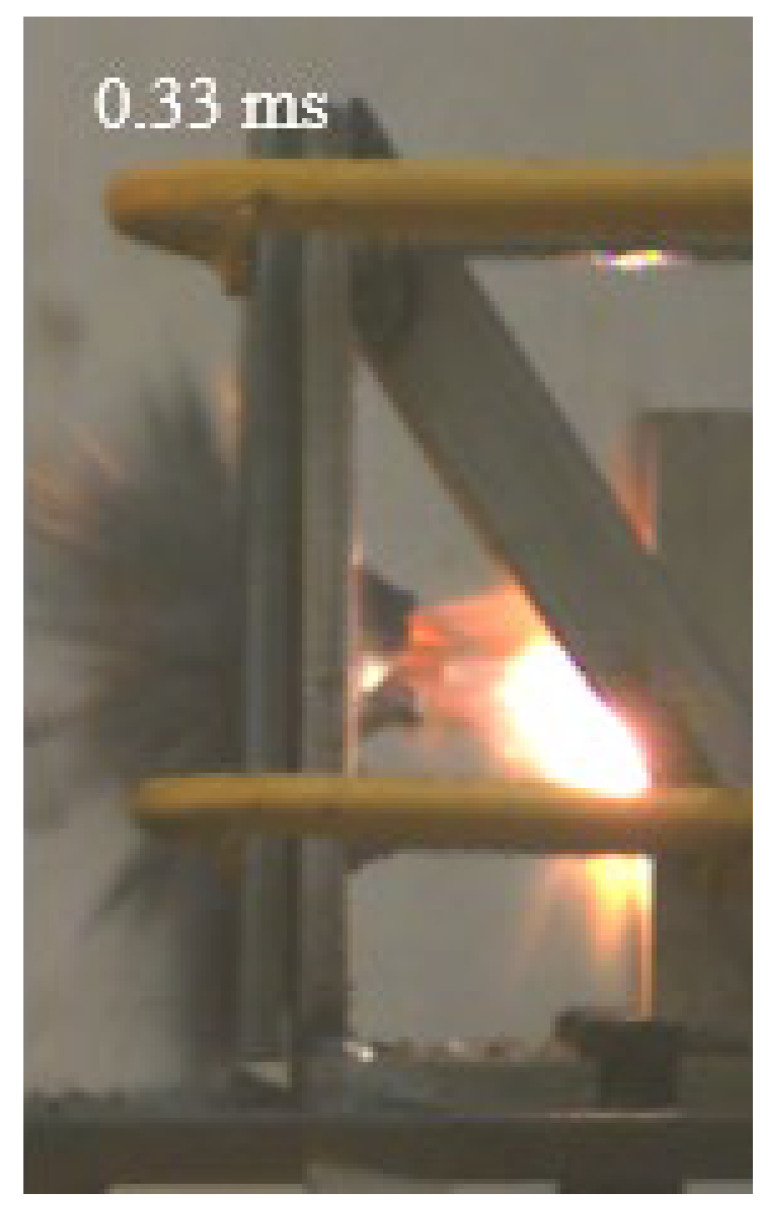	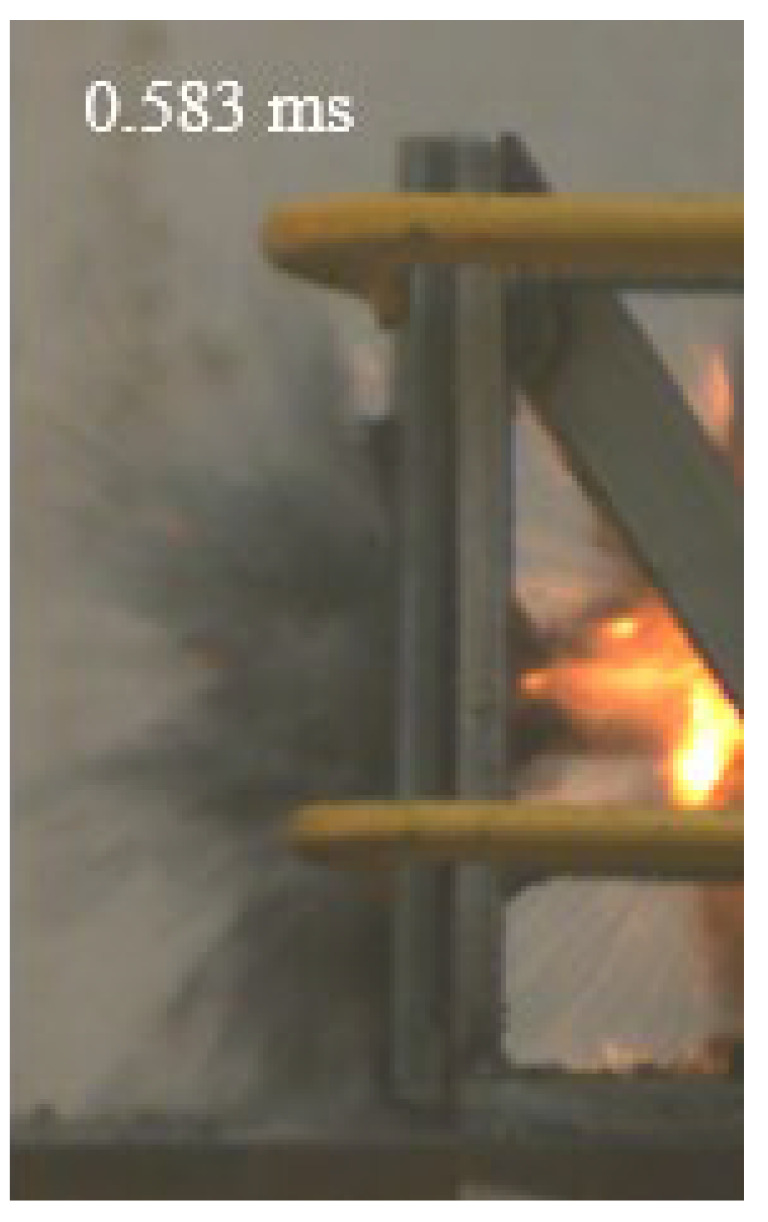	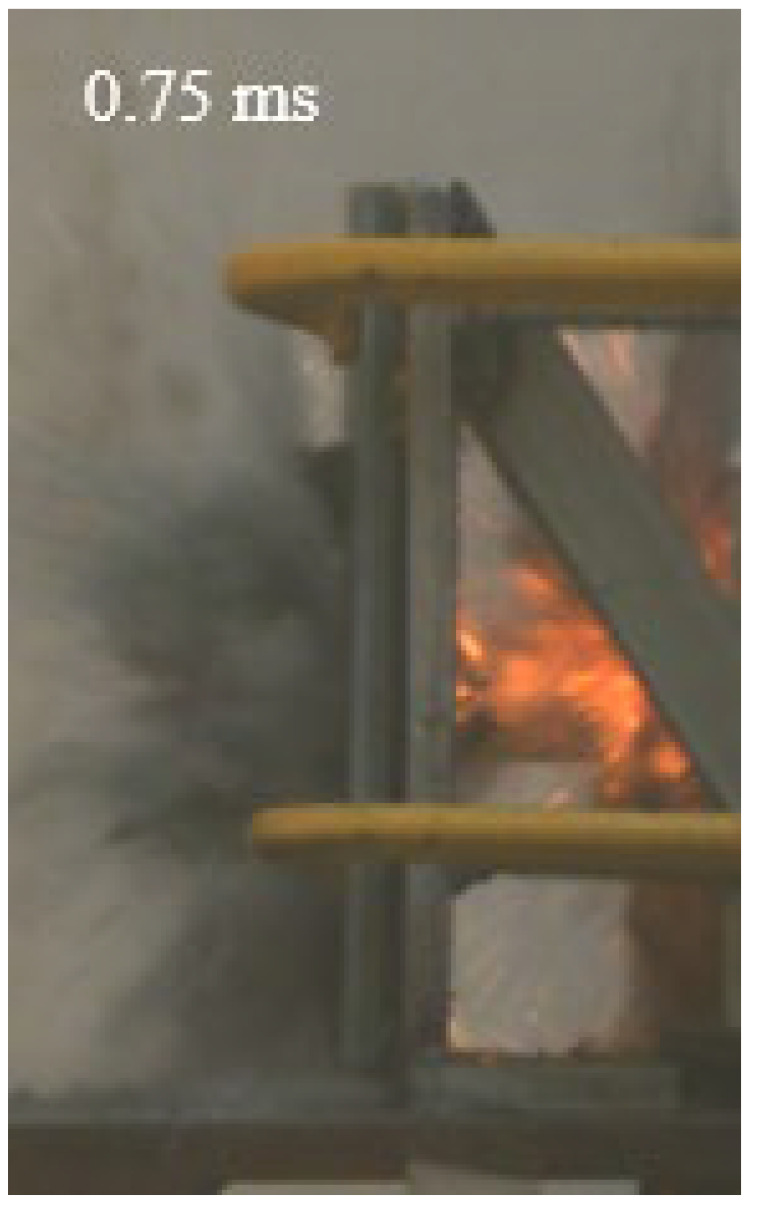
C	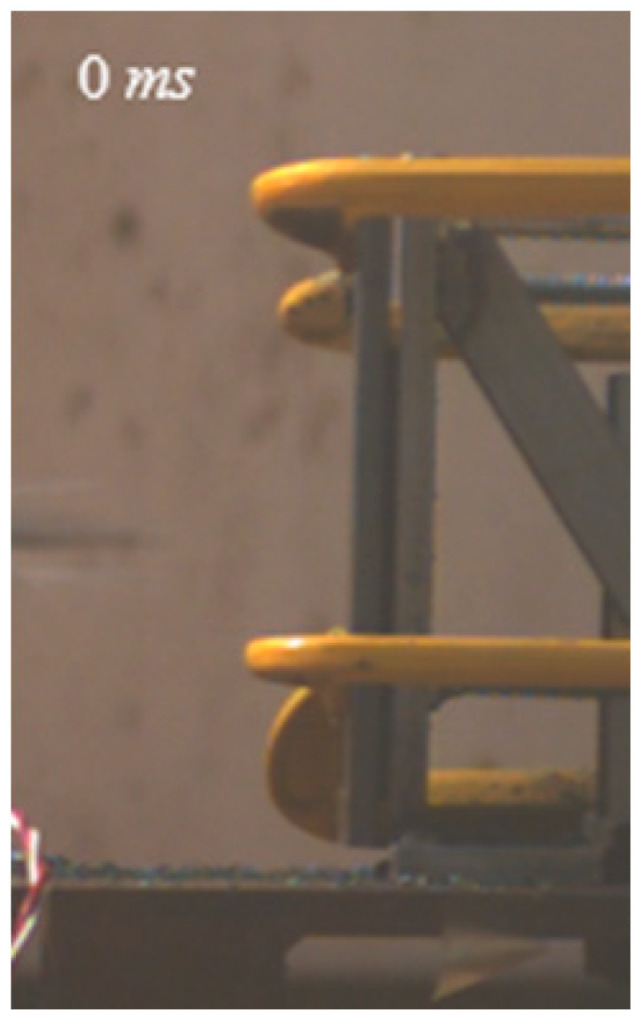	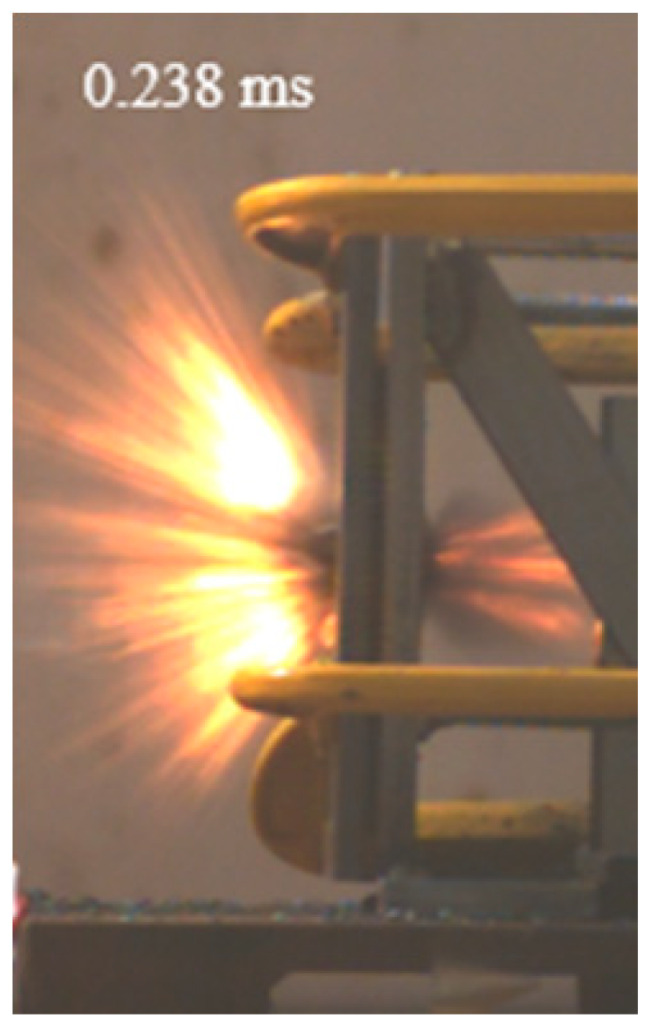	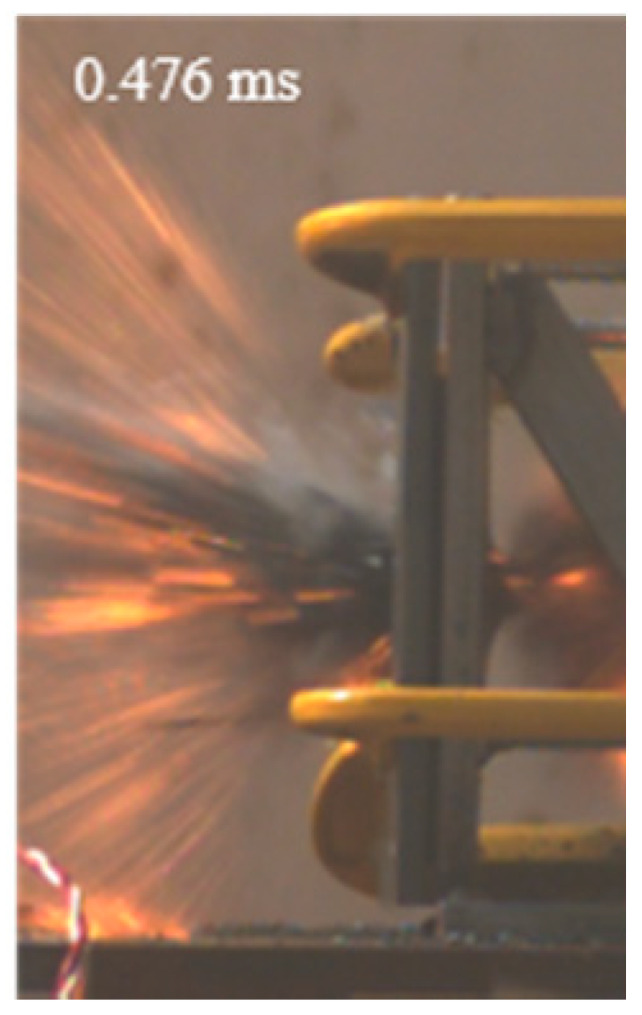	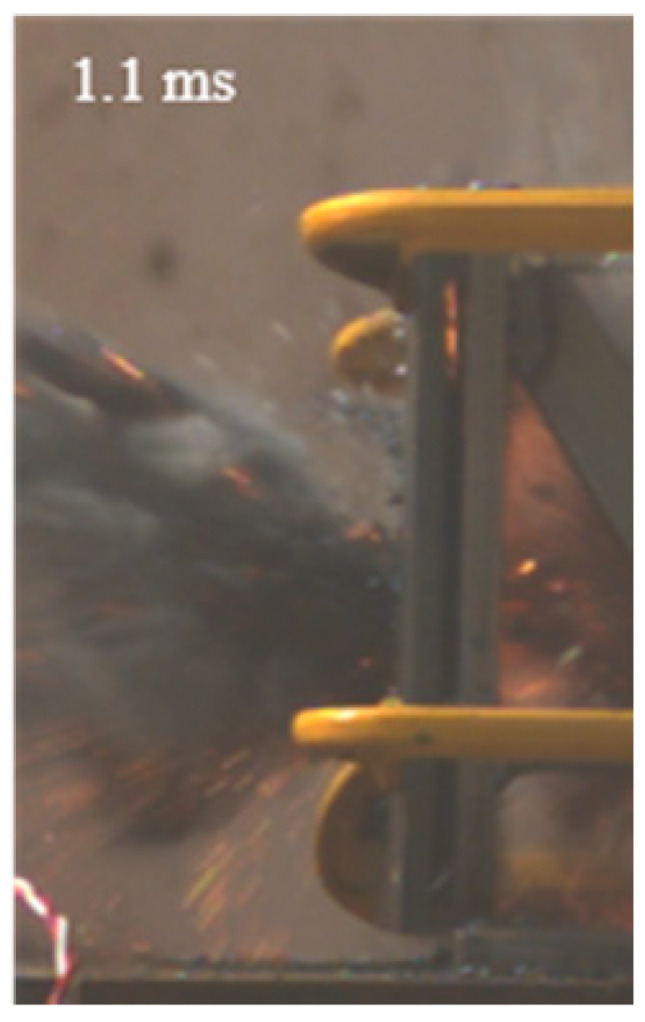	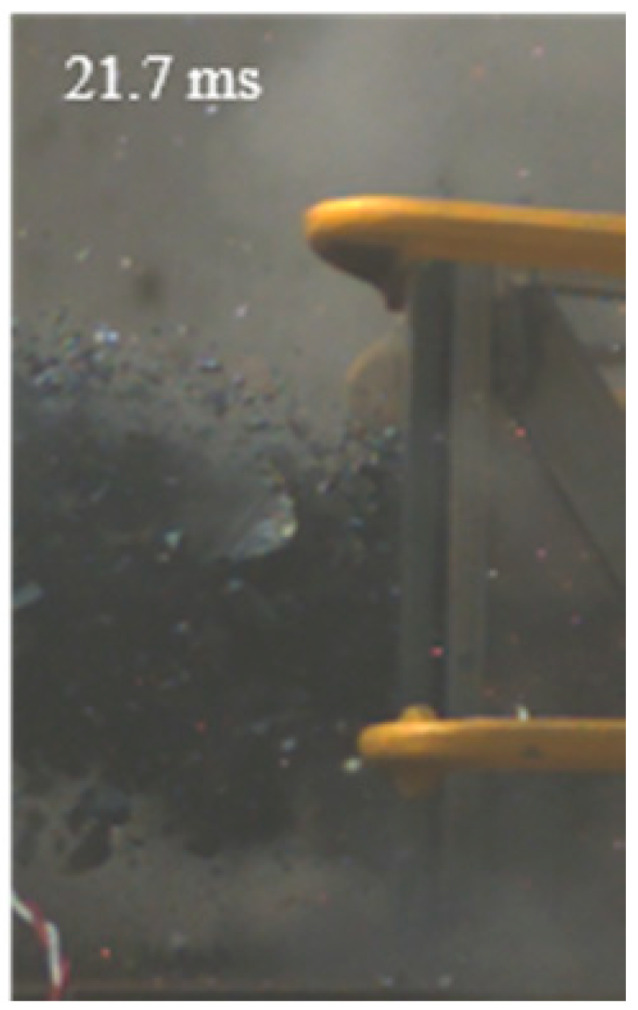

**Table 5 materials-15-03918-t005:** Damage characteristics of the targets after penetration.

Configuration of the Targets	F	M	B	C
Front face	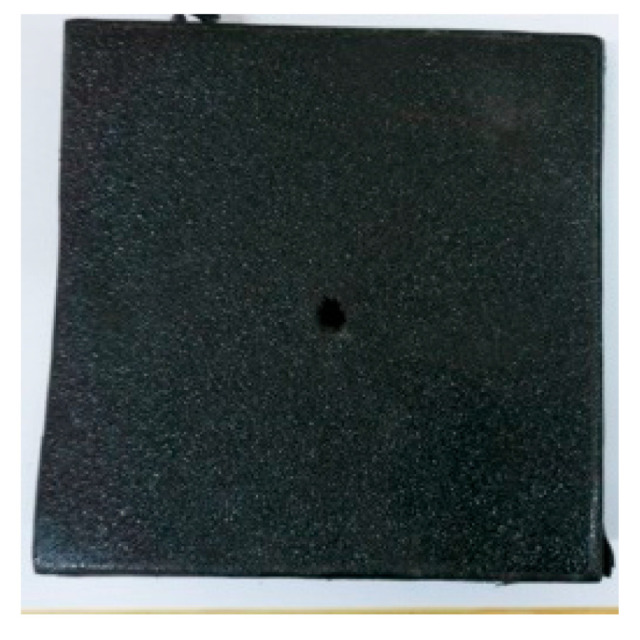	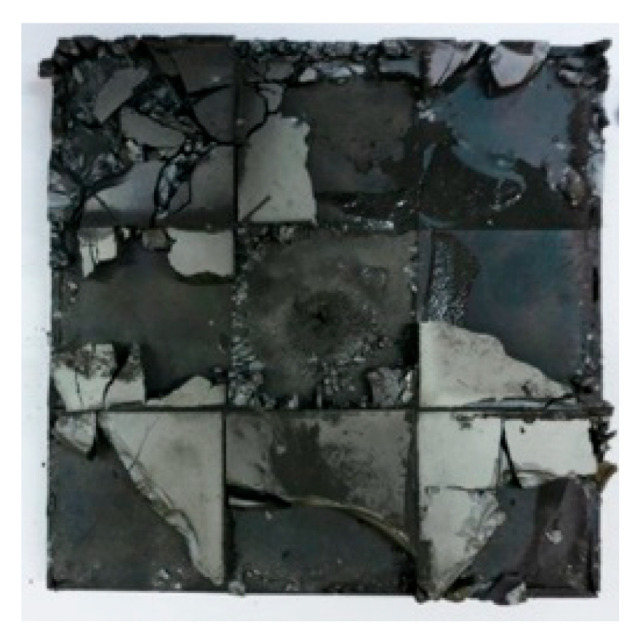	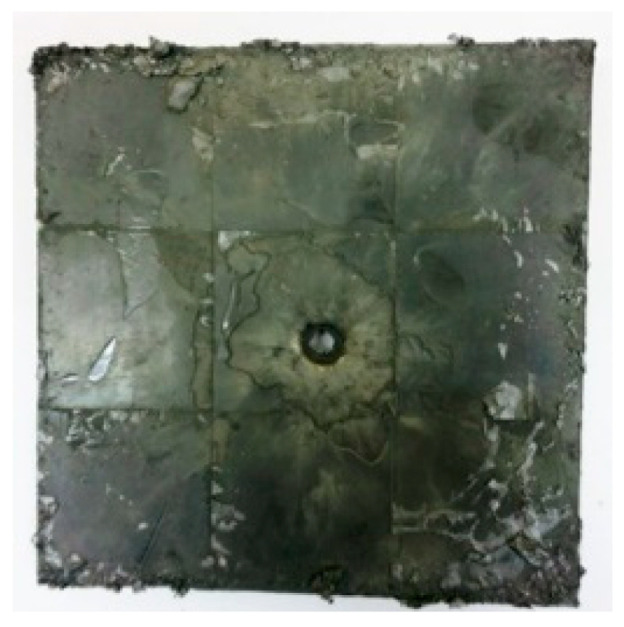	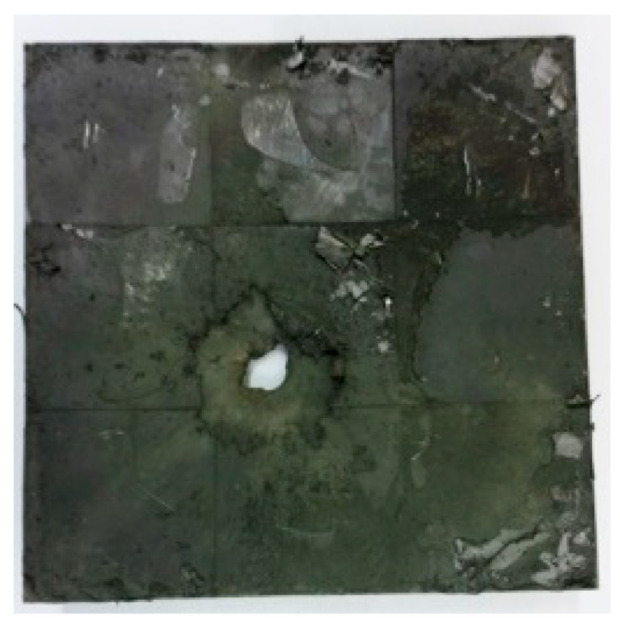
Rear face	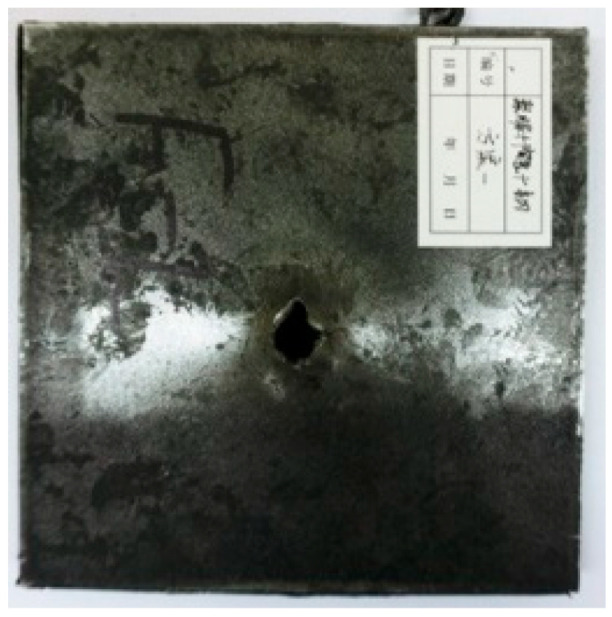	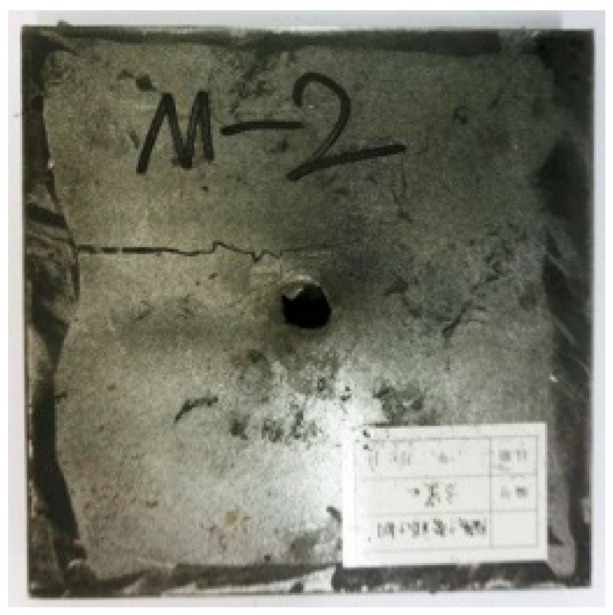	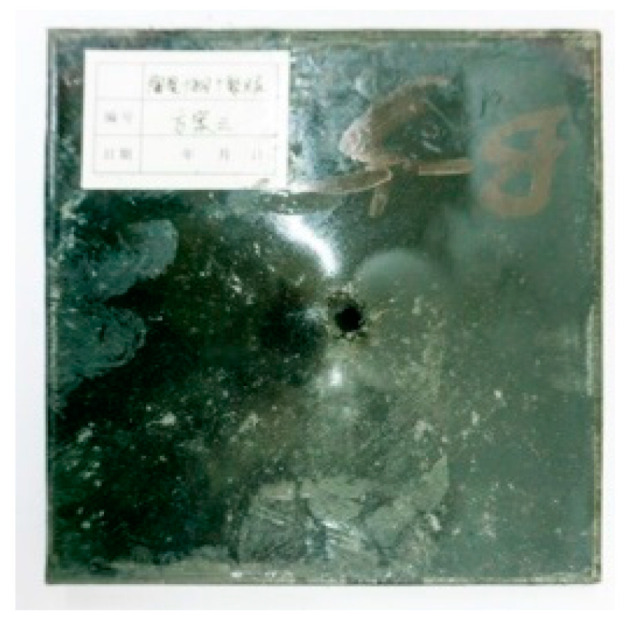	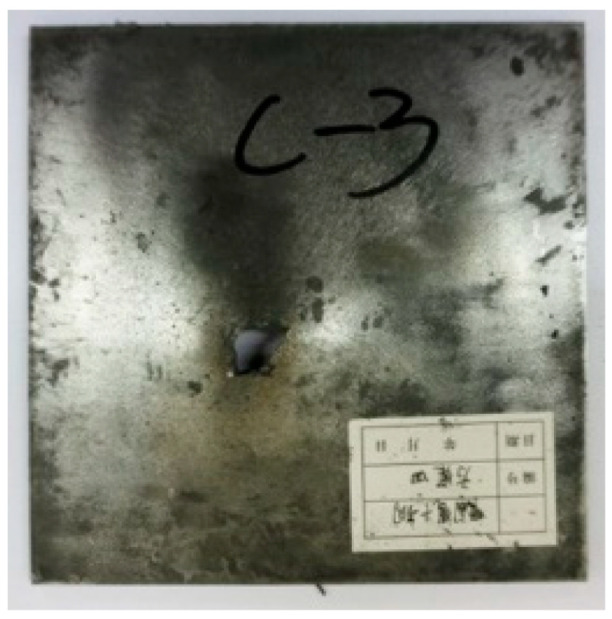

**Table 6 materials-15-03918-t006:** Perforation of the targets.

Configuration Template	F3	M3	B1	B5	B2
Damage Patterns	a. Shearing-Hole	b. Self-Healing	c. Spallation	d. Perforation	e. Cracking
Front face	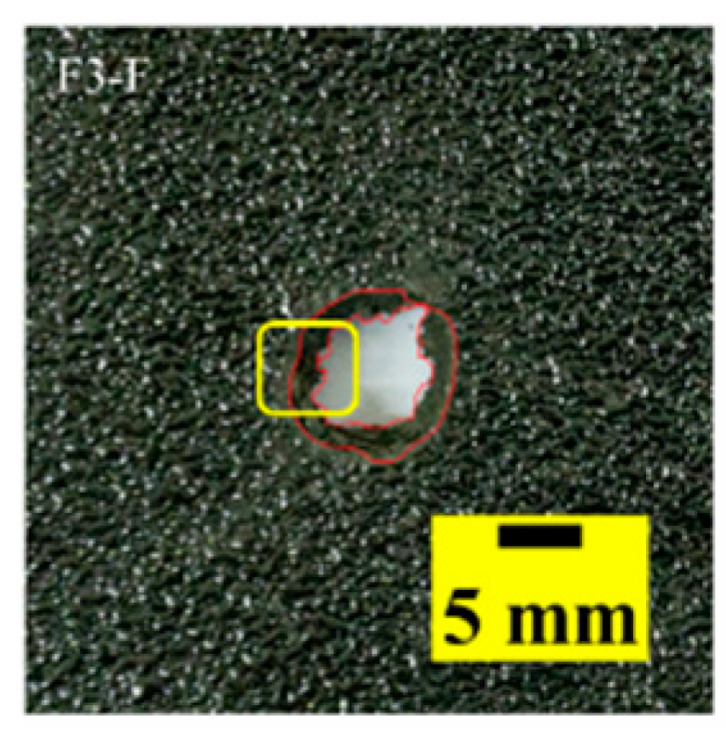	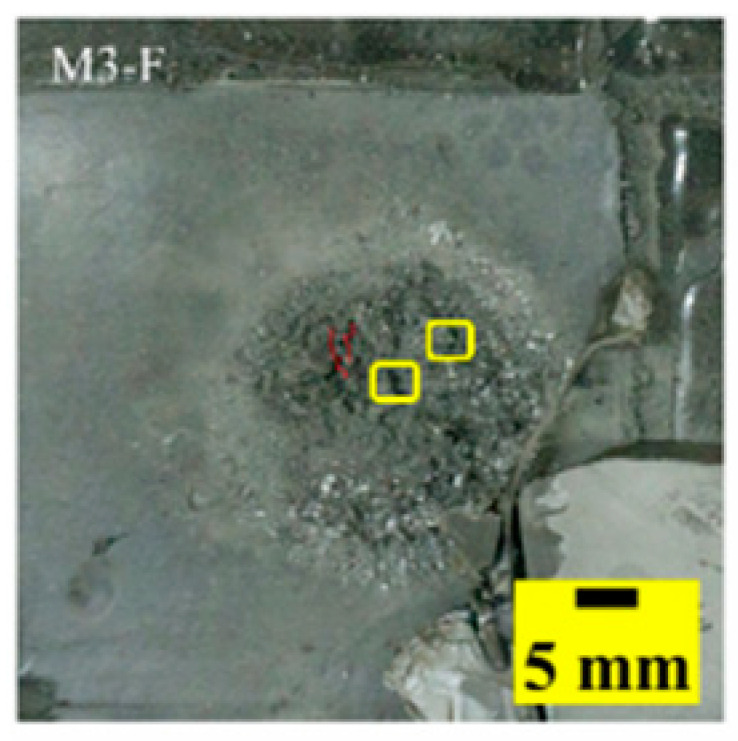	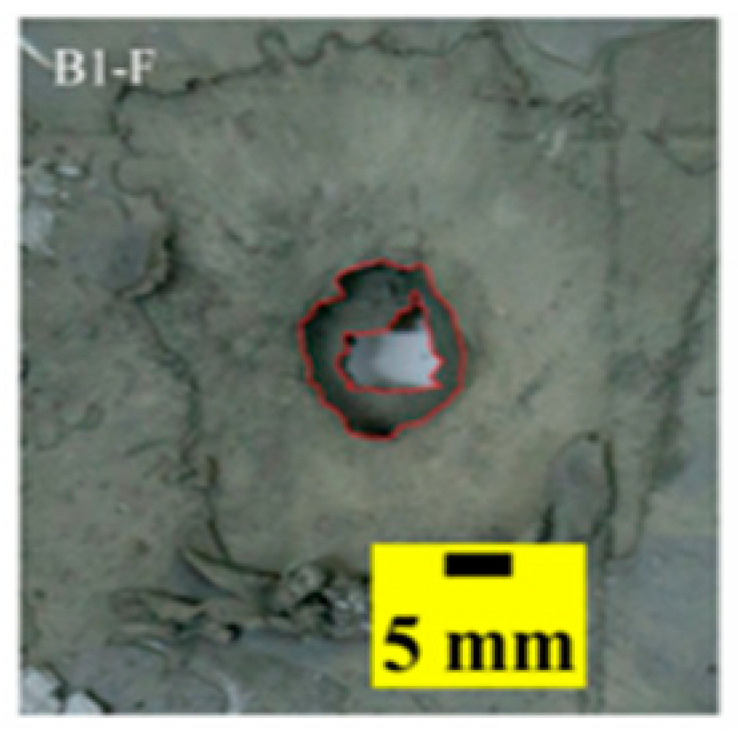	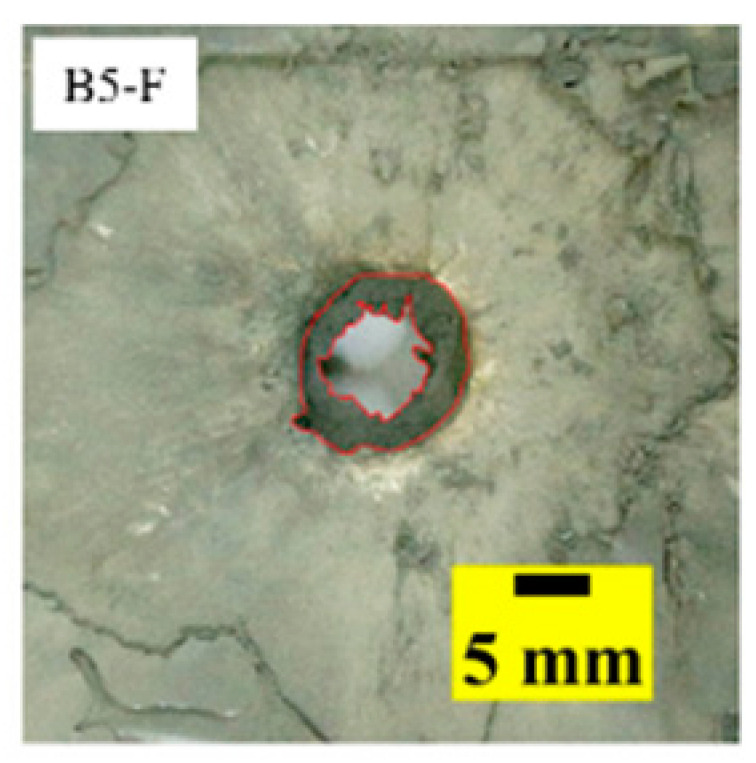	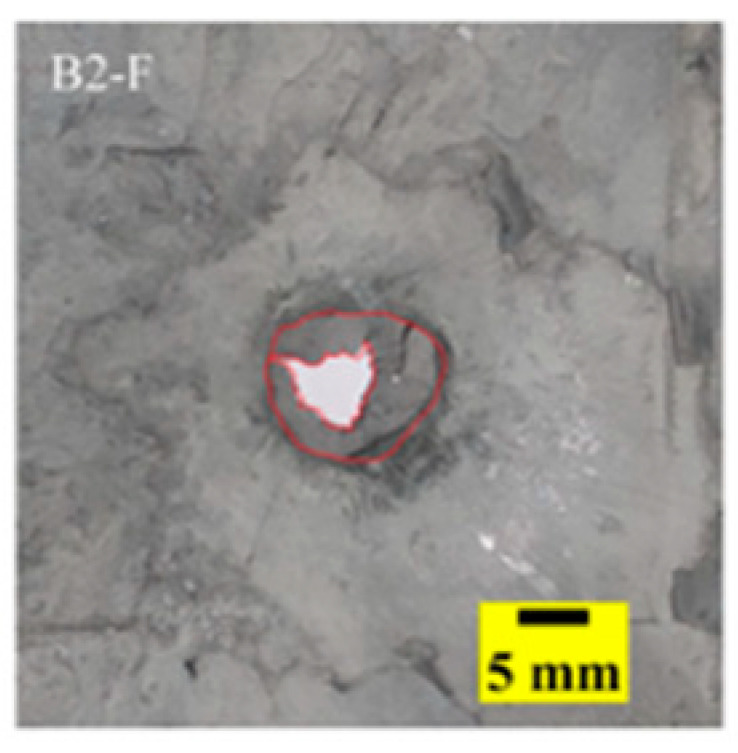
Rear face	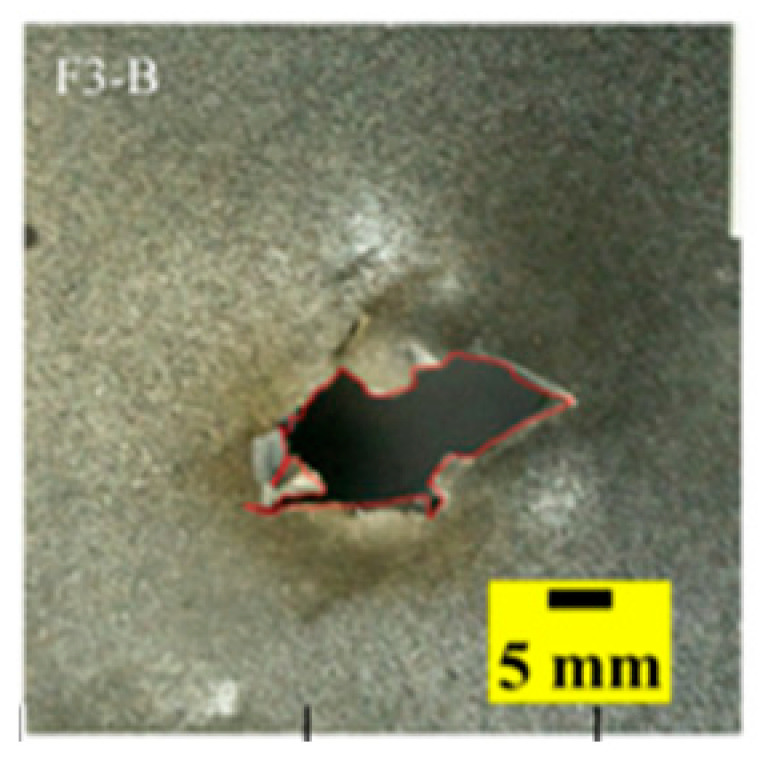	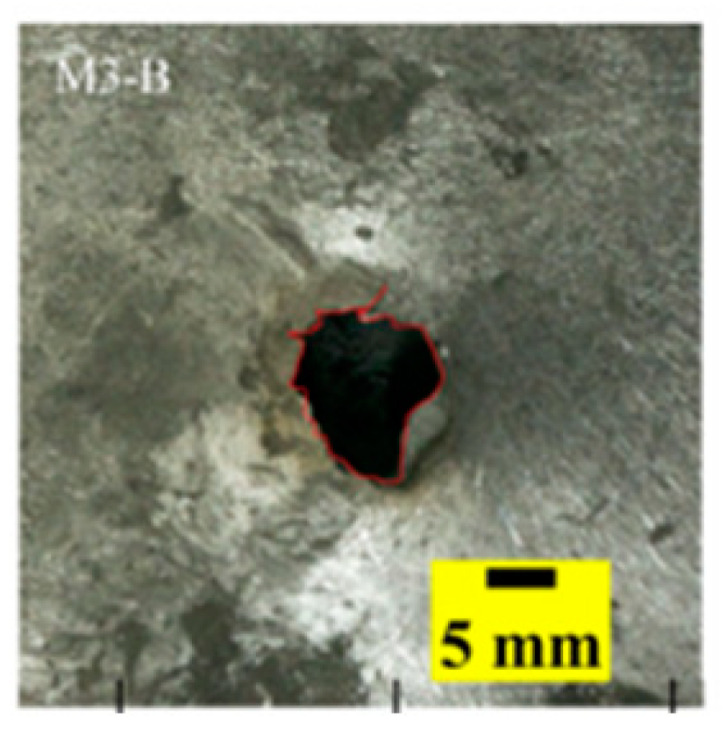	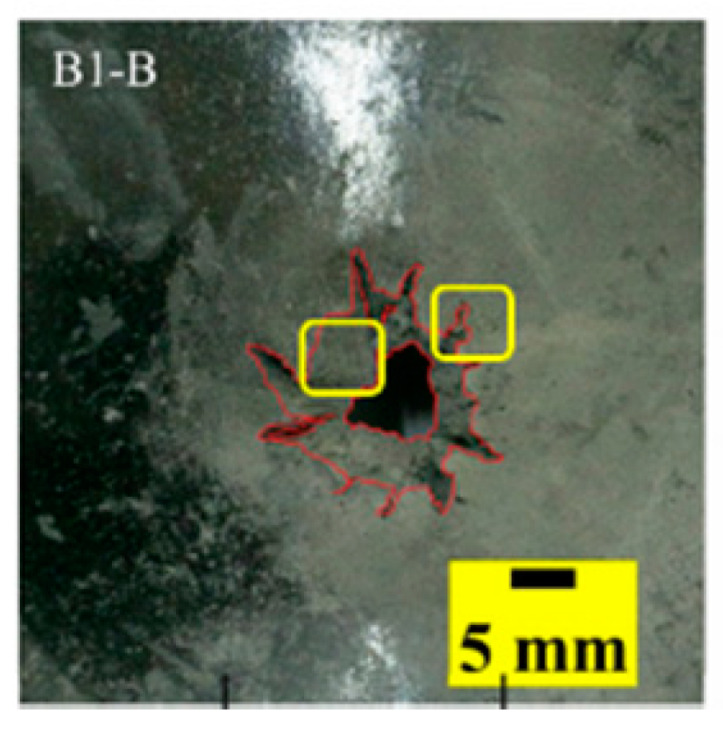	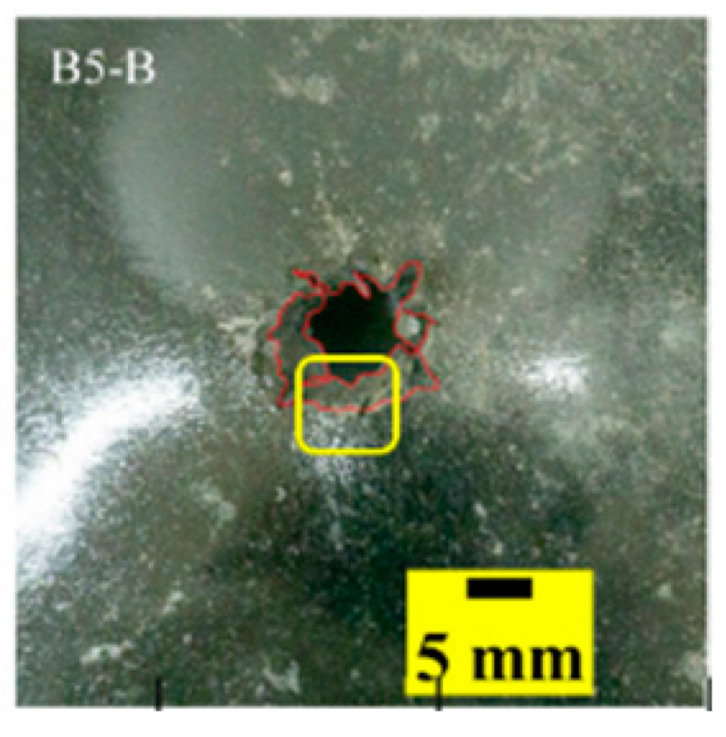	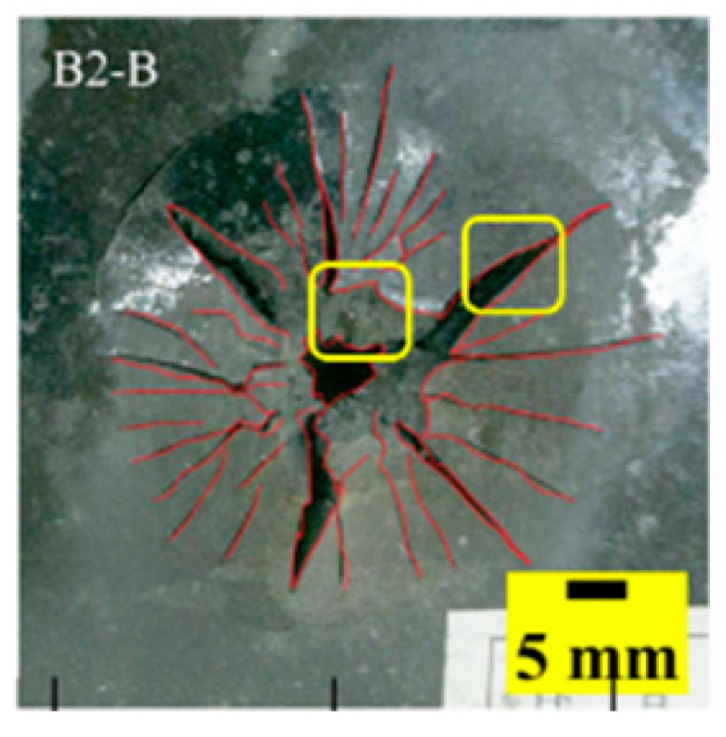

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
