# Peer review of "Ballistic Performance of Polyurea-Reinforced Ceramic/Metal Armor Subjected to Projectile Impact"

_materials, 2022, doi:10.3390/ma15113918_

Round 1

Reviewer 1 Report

The current study investigates how the use of polyurea coatings/layers affects ballistic performance of composite (ceramic-steel) armors. The authors carried out ballistic impact experiments for different target configurations. The tests were recorded with high-speed camera. After the tests the damaged targets were investigated in order to determine the failure patterns and the mechanisms of projectile/target interaction. The scanning electron microscopy (SEM) was used for observation of the microstructure of the penetration craters and targets fracture morphologies. The subject of the paper is interesting, especially for engineers who are looking for new methods of increasing of armors protection capabilities. The proper methodology was adopted and valuable results were obtained. However, there are some minor/major issues that should be explained/corrected before the publication of the paper. The following problems need to be discussed or explained by the author for better understanding:

GENERAL COMMENTS

  1. The subject of the paper is of current interest and in accordance with the profile of the "MATERIALS" journal.
  2. The authors used adequate research methods and tools, however the presentation of the results can be improved, e.g. the recordings from the high speed camera. There is only one frame (picture) presented for each of the target variants (Table 4) what significantly reduces the amount of information that could be collected from the tests. Please add more frames in order to show the course of the penetration/perforation processes.
  3. There is a significant problem with references (Figures, Tables) in this article. The authors probably used automatic algorithm of references linking. However, in the text of the pdf version of the article instead of reference numbers the following message appears: Error! Reference source not found. For this reason it is impossible to evaluate if the references were adopted correctly and whether all of the figures or tables were mentioned in the text.
  4. Authors need to pay more attention to editing the article, e.g. the numbering of chapters. Subchapters in the second chapter should be numbered 2.1, 2.2 etc. Numbering style (1), (2) is used for equations.
  5. The English language, style and grammar must be improved within the whole text. Please use simple sentences that are easier to understand.
  6. In the article the influence of polyurea coating on the fracture intensity in terms of multi-hit resistance of ceramic layers was completely neglected. It seems that this effect is crucial in increasing the ballistic performance of such armors. Please explain.

ABSTRACT

  1. “…rate dependent feature…” please be more specific, did you mean strain hardening? Please explain.
  2. Bi-layer ceramic armour in my understanding defines an armour in which two layers of ceramics are used. In the article a ceramic/metal armour was analyzed (composite containing two layers made of different materials). Please correct the nomenclature in the whole text.
  3. “The results showed that the mass efficiency of polyurea-coated armor was higher than that…” please give some numbers, maybe the increase of efficiency in %?
  4. Please consider reviewing the abstract in terms of highlighting the motivation and the novelty of the study.

INTRODUCTION

  1. It is suggested that the order of writing in the introduction should be revised according to the summary of previous work, existing problems, and then to the order of the main content of this paper.
  2. “…ballistic performance of ceramic armor is greatly influenced by the type of armor ceramics caused by their various material properties…” this is obvious, please avoid such statements.
  3. “…ballistic performance can be significantly improved by setting a reasonable structure.” Please be more specific, add some examples, avoid shorthands.
  4. “Hence, it is necessary to explore the ballistic performance of the novel armor system containing multi-layers by experimental methodology.” Experiments can be supported with numerical simulations.
  5. “performance of ceramic composite armor” should be: composite armors containing, among others, ceramic layers.
  6. “Meanwhile, the ballistic performance of ceramic composite armor has been improved by introducing new materials, such as Kevlar and UHMWPE[23-25] . Such composite armors may achieve considerable ballistic resistance. However, they are too costly and complicated to install.” It is not true. They are not so expensive, and mounting is very similar.
  7. “Therefore, the ballistic performance of composite armor containing novel materials is required.” Looks like missing word, maybe increasing or evaluating of ballistic performance…

2. EXPERIMENTAL MATERIALS AND TARGET CONFIGURATION

  1. “The projectile was made of tungsten alloy formed into the shape of a cylinder with 8 mm diameter and 30 mm length.” What was the motivation of using of such not standardised projectile? Now it is difficult to relate the areal density of targets to the normalized levels of protection, e.g. according to STANAG 4569 agreement. Please explain.

4. EXPERIMENTAL RESULTS

  1. “It implies that the bulletproof performance of polyurea reinforced ceramic armor was is not weaker than that of bi-layer ceramic armor.” was or is?

Author Response

Response to reviewer

Dear reviewer,

The authors would like to thank you for the constructive suggestions and insightful comments concerning our manuscript entitled ‘Ballistic Performance of Polyurea Reinforced Ceramic/Metal Armor Subjected to Projectile Impact’ (materials- 1747726). Those comments and suggestions are all valuable and very helpful for revising and improving our paper, as well as the important guiding significance to our research. We have studied the comments and suggestions carefully and the responses to your comments and suggestions are listed below this letter in highlight mark. In addition, the main revised portions that respond to reviewers' comments are also marked using highlight mark in the manuscript.

We hope that the revised version of the manuscript can meet your requirements and the level of scientific research papers.

I look forward to hearing from you soon.

With the best wishes,

Yours sincerely,

All authors

Replies to your comments:

We shall answer your comments one by one here in order to respond to them more clearly.

  1. The authors used adequate research methods and tools, however the presentation of the results can be improved, e.g. the recordings from the high speed camera. There is only one frame (picture) presented for each of the target variants (Table 4) what significantly reduces the amount of information that could be collected from the tests. Please add more frames in order to show the course of the penetration/perforation processes.

Response 1: Thank you for your detailed suggestion. We have added more frames in

Table 4.

Tab. 4 High-speed photography of the targets

Configuration template

Moment 1

Moment 2

Moment 3

Moment 4

Moment 5

F

  1. There is a significant problem with references (Figures, Tables) in this article. The authors probably used automatic algorithm of references linking. However, in the text of the pdf version of the article instead of reference numbers the following message appears: Error! Reference source not found. For this reason it is impossible to evaluate if the references were adopted correctly and whether all of the figures or tables were mentioned in the text.
  2. Authors need to pay more attention to editing the article, e.g. the numbering of chapters. Subchapters in the second chapter should be numbered 2.1, 2.2 etc. Numbering style (1), (2) is used for equations.

Response 2 & 3: We are really sorry for such a stupid mistake in the manuscript and thank you for raising such a detailed suggestion. We have carefully examined and corrected the mistake.

  1. The English language, style and grammar must be improved within the whole text. Please use simple sentences that are easier to understand.

Response 4: Many thanks for your kind comments and valuable suggestions. We had retouched the language before submission, and have improved the grammar & writing style of the manuscript after review.

  1. In the article the influence of polyurea coating on the fracture intensity in terms of multi-hit resistance of ceramic layers was completely neglected. It seems that this effect is crucial in increasing the ballistic performance of such armors. Please explain.

Response 5: Thank you for your professional suggestion. In the present study, we aim to investigate the ballistic performance of polyurea reinforced ceramic/metal armor by the proposed method. The fracture strength of polyurea on ballistic performance will be considered in the follow-up study, and the multi-hit resistance of ceramic layers will be studied.

  1. “…rate dependent feature…” please be more specific, did you mean strain hardening? Please explain.

Response 6: Thank you for your professional suggestion. The strength of polyurea increases with the increase of strain rate, which is called strain rate effect. The author has revised “rate dependent feature” to “dynamic feature”.

  1. Bi-layer ceramic armour in my understanding defines an armour in which two layers of ceramics are used. In the article a ceramic/metal armour was analyzed (composite containing two layers made of different materials). Please correct the nomenclature in the whole text.
  2. “The results showed that the mass efficiency of polyurea-coated armor was higher than that…” please give some numbers, maybe the increase of efficiency in %?
  3. Please consider reviewing the abstract in terms of highlighting the motivation and the novelty of the study.

Response 7, 8 & 9: Thank you for your detailed suggestion. The author has revised as requested in accordance with the requirements of full text.

Revised abstract:Although polyurea attracts extensive attention in impact mitigation due to its protective feature during intensive loading, the ballistic performance of polyurea reinforced ceramic/metal armor remains unclear. In the present study, polyurea reinforced ceramic/metal armor with different structures was designed, including three kinds of coating positions of polyurea. The ballistic tests were conducted with the ballistic gun subjected to the tungsten projectile formed as a cylinder with 8 mm diameter and 30 mm length, and the deformation process of the tested targets was recorded with a high-speed camera. The ballistic performance of the polyurea reinforced ceramic/metal armor was evaluated by the mass efficiency. The damaged targets were investigated in order to determine the failure patterns and the mechanisms of interaction between projectile and target. The scanning electron microscope (SEM) was used to observe the microstructure of polyurea to understand the failure mechanisms of polyurea. The results showed that the mass efficiency of polyurea-coated armor was 89% higher than that of ceramic/metal armor, which implies that polyurea-coated ceramic armor achieved higher ballistic performance by lighter mass quality than that of ceramic/metal armor. The improvement of ballistic performance was due to the energy absorbed by polyurea during glass transition. These results are promising for further applications of polyurea reinforced ceramic/metal armor.

  1. It is suggested that the order of writing in the introduction should be revised according to the summary of previous work, existing problems, and then to the order of the main content of this paper.

Response 10: Thank you for your professional suggestion. The author has revised the introduction as requested

  1. “…ballistic performance of ceramic armor is greatly influenced by the type of armor ceramics caused by their various material properties…” this is obvious, please avoid such statements.

Response 11: Thank you for your professional suggestion. The author has revised as requested.

  1. “…ballistic performance can be significantly improved by setting a reasonable structure.” Please be more specific, add some examples, avoid shorthands.

Response 12: Thank you for your professional suggestion. The author has revised as requested.

“In addition to material properties, the ballistic performance can be significantly improved by setting a reasonable structure[11-15]. For example, the ballistic limit can be increased by adjusting the thickness ratio and the restraint form of the ceramic block.”

  1. “Hence, it is necessary to explore the ballistic performance of the novel armor system containing multi-layers by experimental methodology.” Experiments can be supported with numerical simulations.

Response 13: Thank you for your professional suggestion. The accuracy of numerical simulation results depends on the material model and parameters. The existing numerical models are difficult to accurately predict the dynamic failure characteristics of ceramic and polyurea. In order to obtain the appropriate model and parameters, the ballistic experiment is conducted. The author will use numerical simulations to optimize the structure and analyze the parameters of polyurea reinforced ceramic/armor in the following research.

  1. “performance of ceramic composite armor” should be: composite armors containing, among others, ceramic layers.

Response 14: Thank you for your professional suggestion. The author has revised as requested.

  1. “Meanwhile, the ballistic performance of ceramic composite armor has been improved by introducing new materials, such as Kevlar and UHMWPE[23-25] . Such composite armors may achieve considerable ballistic resistance. However, they are too costly and complicated to install.” It is not true. They are not so expensive, and mounting is very similar.

Response 14: Thank you for your professional suggestion. The author has revised as requested. “However, the density of Kevlar fiber was higher, and the melting point of UHMWPE was lower, which limited the application of the two materials.”

  1. “Therefore, the ballistic performance of composite armor containing novel materials is required.” Looks like missing word, maybe increasing or evaluating of ballistic performance…

Response 16: Thank you for your detailed suggestion. The author has revised as requested.

  1. “The projectile was made of tungsten alloy formed into the shape of a cylinder with 8 mm diameter and 30 mm length.” What was the motivation of using of such not standardised projectile? Now it is difficult to relate the areal density of targets to the normalized levels of protection, e.g. according to STANAG 4569 agreement. Please explain.

Response 17: Thank you for your detailed suggestion. The authors then explain the relationship between our results and previous results. The author listed detailed parameter properties and chemical components in Tab. 1 and Tab. 2 of the manuscript. The shape of the projectile body is greatly simplified in order to exclude the influence of other factors on the protective performance. The authors will consider the influence of different projectile shapes and establish the ballistic limit theory model of composite armor in subsequent research.

  1. “It implies that the bulletproof performance of polyurea reinforced ceramic armor was is not weaker than that of bi-layer ceramic armor.” was or is?

Response 18: We are really sorry for such a stupid mistake in the manuscript and thank you for raising such a detailed suggestion. This sentence has been revised to “It implied that the bulletproof performance of polyurea reinforced ceramic armor is not weaker than that of ceramic/metal armor”

Thank you again for the constructive suggestions and insightful comments.

We hope that the revised version of the manuscript can meet your requirements and the level of scientific research papers.

I look forward to hearing from you soon.

With the best wishes,

Yours sincerely,

All authors

Reviewer 2 Report

The paper considers a rather original approach to the manufacture of multilayer materials for use as ballistic protection. However, reading it raises many questions. 

1) Firstly, for that journal as "Materials", as I understand it, it is very important to pay attention to the features of the internal structure of the materials under study (i.e. their microstructure, phase composition, texture, etc.) both in the initial state and after any processing/testing, etc. In other words, it is necessary to establish/show exactly how these factors affect certain physical and other properties of the material. I did not find anything like this in this manuscript! 

2) Secondly, the authors did not even show a single option when the material they proposed completely survived in the case of ballistic tests. This makes the value of the result obtained rather doubtful. 

3) Returning to the Introduction, and also regarding the Discussion of the results. The authors lightly mention such widely used materials as steel and aluminum alloys used as armor. However, titanium alloys and metal-matrix composites based on them are not mentioned in a word, which have shown their superiority over many other armor materials (see, for example, https://doi.org/10.1007/978-94-024-2021-0_13, or https://doi.org/10.15407/ufm.20.02.052).  I believe that without a correct comparison of the ballistic resistance of the material proposed by the authors with other known analogues, it will be difficult for readers to assess the quality of the proposed article. 

There are also comments on the design of the manuscript (few examples only): 

- Table 4 is not informative and most likely it is not a table, but a figure. I think that it can be painlessly removed. 

- When analyzing Fig. 9 the question arises (absolutely brittle destruction of polyurea) - where is the plasticity of 400% declared by the authors at the very beginning? 

- What is Fig. 1 inside Fig. 9? There must be some kind of unified system of notation, and not confusion!

Author Response

Response to reviewer

Dear reviewer,

The authors would like to thank you for the constructive suggestions and insightful comments concerning our manuscript entitled ‘Ballistic Performance of Polyurea Reinforced Ceramic/Metal Armor Subjected to Projectile Impact’ (materials- 1747726). Those comments and suggestions are all valuable and very helpful for revising and improving our paper, as well as the important guiding significance to our research. We have studied the comments and suggestions carefully and the responses to your comments and suggestions are listed below this letter in highlight mark. In addition, the main revised portions that respond to reviewers' comments are also marked using highlight mark in the manuscript.

We hope that the revised version of the manuscript can meet your requirements and the level of scientific research papers.

I look forward to hearing from you soon.

With the best wishes,

Yours sincerely,

All authors

Replies to your comments:

We shall answer your comments one by one here in order to respond to them more clearly.

  1. Firstly, for that journal as "Materials", as I understand it, it is very important to pay attention to the features of the internal structure of the materials under study (i.e. their microstructure, phase composition, texture, etc.) both in the initial state and after any processing/testing, etc. In other words, it is necessary to establish/show exactly how these factors affect certain physical and other properties of the material. I did not find anything like this in this manuscript!

Response 1: Many thanks for your kind comments and valuable suggestions. We have added the SEM image of initial state of polyurea. The current study investigates how the use of polyurea coatings/layers affects ballistic performance of ceramic/steel armors. We carried out ballistic impact experiments for different target configurations. The tests were recorded with high-speed camera. After the tests the damaged targets were investigated in order to determine the failure patterns and the mechanisms of projectile/target interaction. The scanning electron microscopy (SEM) was used for observation of the microstructure of the penetration craters and targets fracture morphologies. We consider that the subject of the paper is in accordance with the profile of the "MATERIALS" journal

(a). Surface area

(b) Initial state of polyurea

(c). Fracture area debris

Fig. 7 SEM images of polyurea on the target of configuration F and in the initial state

  1. Secondly, the authors did not even show a single option when the material they proposed completely survived in the case of ballistic tests. This makes the value of the result obtained rather doubtful.

Response 2: Thank you for your professional suggestion. The authors then explain the rationality of the experimental design and the authenticity of the results. We use the residual depth of the penetration to measure the ballistic performance of the composite armor. In the experiment, the speed of the bomb body is large enough to pass through the target. We ensure the authenticity of the experimental results and can upload all experimental data and photos.

  1. Returning to the Introduction, and also regarding the Discussion of the results. The authors lightly mention such widely used materials as steel and aluminum alloys used as armor. However, titanium alloys and metal-matrix composites based on them are not mentioned in a word, which have shown their superiority over many other armor materials (see, for example, https://doi.org/10.1007/978-94-024-2021-0_13, or https://doi.org/10.15407/ufm.20.02.052). I believe that without a correct comparison of the ballistic resistance of the material proposed by the authors with other known analogues, it will be difficult for readers to assess the quality of the proposed article.

Response 3: Thank you for your professional suggestion. We carefully read the literature recommended by the reviewer and made a correct citation (https://doi.org/10.1007/978-94-024-2021-0_13). As stated in the introduction, this paper refers to a large number of research results. The authors then explain the relationship between our results and previous results. We aim to study the effect of polyurea coating on ceramic/metal armor. We chose the armor steel, which is widely used in most military equipment. The author listed detailed parameter properties and chemical components in Tab. 1 and Tab. 2 of the manuscript. The tensile strength of armor steel is 1700, which is much higher than that of titanium alloy. Therefore, readers can compare it with other research results.

Tab. 1 Material properties

Material

Material properties

Polyurea

Density (kg/m³)

Solid content

Tensile strength (MPa)

Tearing strength (MPa)

Elongation

Solidification time (s)

1010

≥96%

10

≥40KN/m

400%

45

SiC ceramic

Density (kg/m³)

Vickers hardness (MPa)

Bending strength (MPa)

Compressive strength (MPa)

Crystal density (μm)

Elasticity modulus (GPa)

3130

2600

400

2200

5

430

Armor steel

Density (kg/m³)

Brinell hardness (HB)

Yeld strength (MPa)

Tensile strength (MPa)

Elongation

7850

500

1400

1700

10%

Aluminum 6061-T6

Density (kg/m³)

Tensile strength (MPa)

Yield strength (MPa)

Elongation (%)

2850

318

257

9.9

Tungsten alloy

Density (kg/m³)

Yield strength (MPa)

Elongation (%)

Rockwell hardness (HRC)

17600

742

8.8

27

Tab. 2 Chemical composition of materials. (in wt.%)

Material

Chemical composition

Armor steel

C*

Si*

Mn*

P

S

Cr*

Ni

Mo*

B*

0.32

0.4

1.2

0.01

0.003

1.0

1.8

0.7

0.005

Aluminium 6061-T6

SI

Fe

Cu

Mn

Mg

Cr

Zn

Ti

Al

0.59

0.369

0.246

0.063

1.025

0.201

0.103

0.028

97.37

Tungsten alloy

W

Ni

Fe

93

5.1

1.9

  1. Table 4 is not informative and most likely it is not a table, but a figure. I think that it can be painlessly removed.

Response 4: Thank you for your professional suggestion. We complete the images of the penetration process to provide the reader with more information. The process of projectile penetrating the target is described and analyzed.

Tab. 4 High-speed photography of the targets

Configuration template

Moment 1

Moment 2

Moment 3

Moment 4

Moment 5

F

  1. When analyzing Fig. 9 the question arises (absolutely brittle destruction of polyurea) - where is the plasticity of 400% declared by the authors at the very beginning?

Response 5: Thank you for your professional suggestion. The authors then explain the rationality of this result. The polyurea has good ductility in quasi-static state (Usually around 10-2 s-1), the elongation of which is 400% (One of the plastic properties of materials). Measuring material elongation due to axial force is usually carried out by a standard tensile strength test. A strip or rod with a certain length and a uniform cross-sectional area, fixed at one end, is subjected to a tensile load along the specimen’s axis in quasi-static state. When the strain rate increases, most materials exhibit a strain rate dependent feature. The glass transition of polyurea is occurred under intensive loading (such as blast and penetration), which makes the brittle failure of polyurea accompanied by energy absorption. This property enables polyurea to be used in explosion and penetration protection. In this study, this transition was occurred in the polyurea in configuration F, which explained the excellent protection performance of configuration F.

  1. What is Fig. 1 inside Fig. 9? There must be some kind of unified system of notation, and not confusion!

Response 6: We are really sorry for such a stupid mistake in the manuscript and thank you for raising such a detailed suggestion. There must be some problems with the software of the automatic algorithm of references linking. We have corrected the unified system of notation and uploaded the PDF version.

(a) Debris and holes

(b) Brittle crack and holes

(c) Cacks and holes

Fig. 9 SEM images of polyurea on the target of configuration B1

Thank you again for the constructive suggestions and insightful comments.

We hope that the revised version of the manuscript can meet your requirements and the level of scientific research papers.

I look forward to hearing from you soon.

With the best wishes,

Yours sincerely,

All authors

Round 2

Reviewer 2 Report

The authors partially took into account my comments. However, since I did not see a full response to my remark regarding the lack of data on the evolution of the microstructure and phase composition of ALL the constituent parts of the "sandwiches" studied in the work, I leave the final decision on accepting the manuscript to the editor's judgment.

Author Response

Response to reviewer

Dear reviewer,

Thank you for your letter and for the reviewers’ comments concerning our manuscript entitled ‘Ballistic Performance of Polyurea Reinforced Ceramic/Metal Armor Subjected to Projectile Impact’ (materials- 1747726). These comments are all valuable and very helpful for revising and improving our paper, as well as the important guiding significance to our researches. We have studied comments carefully and have made corrections which we hope meet with approval.

I look forward to hearing from you soon.

With the best wishes,

Yours sincerely,

All authors

Replies to your comments:

Comments and Suggestions for Authors: The authors partially took into account my comments. However, since I did not see a full response to my remark regarding the lack of data on the evolution of the microstructure and phase composition of ALL the constituent parts of the "sandwiches" studied in the work, I leave the final decision on accepting the manuscript to the editor's judgment.

Response 1: Many thanks for your kind comments and valuable suggestions. The subject of this study is the ballistic performance of polyurea reinforced ceramic/metal armor. We carried out ballistic impact experiments for different target configurations. The results showed that the mass efficiency of polyurea-coated armor was higher than that of bi-layer ceramic armor. A scanning electron microscope was used to observe the microstructure of polyurea and the deformation process of the targets to understand the failure mechanisms of polyurea. These results are promising for further applications of polyurea-coated armor. We consider that the subject of the paper is in accordance with the profile of the "MATERIALS" journal

Thank you again for the constructive suggestions and insightful comments.

We hope that the revised version of the manuscript can meet your requirements and the level of scientific research papers.

I look forward to hearing from you soon.

With the best wishes,

Yours sincerely,

All authors
